



# Explaining apparent particle shrinkage related to new particle formation events in western Saudi Arabia does not require evaporation

Simo Hakala[1], Ville Vakkari[2,3], Heikki Lihavainen[2,4], Antti-Pekka Hyvärinen[2], Kimmo Neitola[1,5], Jenni Kontkanen[1,6], Veli-Matti Kerminen[1], Markku Kulmala[1], Tuukka Petäjä[1], Tareq Hussein[1,7], Mamdouh I. Khoder[8], Mansour A. Alghamdi[9], Pauli Paasonen[1]

[1]Institute for Atmospheric and Earth System Research (INAR) /Physics, Faculty of Science, University of Helsinki, Finland
[2]Finnish Meteorological Institute, Helsinki, Finland
[3]Atmospheric Chemistry Research Group, Chemical Resource Beneficiation, North-West University, Potchefstroom, South Africa
[4]Svalbard Integrated Arctic Earth Observing System (SIOS), Longyearbyen, Norway
[5]Vaisala Oyj, Vantaa, Finland
[6]CSC – IT Center for Science, Espoo, Finland
[7]Environmental and Atmospheric Research Laboratory (EARL), Department of Physics, School of Science, the University of Jordan, Amman 11942, Jordan
[8]Air Pollution Research Department, Environment and Climate Change Research Institute, National Research Centre, El Behooth Str., Dokki, Giza 12622, Egypt.
[9]Department of Environmental Sciences, Faculty of Meteorology, Environment and Arid Land Agriculture, King Abdulaziz University, Jeddah, Saudi Arabia

*Correspondence to*: Simo Hakala (simo.hakala@helsinki.fi)

**Abstract.** The majority of new particle formation (NPF) events observed in Hada al Sham, western Saudi Arabia during 2013-2015, showed an unusual progression where the diameter of a newly formed particle mode clearly started to decrease after the growth phase. Many previous studies refer to this phenomenon as aerosol shrinkage. We will opt to use the term decreasing mode diameter (DMD) event, as shrinkage bears the connotation of reduction in the sizes of individual particles, which does not have to be the case. While several previous studies speculate that ambient DMD events are caused by evaporation of semivolatile species, no concrete evidence has been provided, partly due to the rarity of the DMD events. The frequent occurrence and large number of DMD events in our observations allow us to perform statistically significant comparisons between the DMD and the typical NPF events that undergo continuous growth. In our analysis, we find no clear connection between DMD events and factors that might trigger particle evaporation at the measurement site. Instead, examination of air mass source areas and the horizontal distribution of anthropogenic emissions in the study region leads us to believe that the observed DMD events could be caused by advection of smaller, less-grown, particles to the measurement site after the more-grown ones. Using a Lagrangian single-particle growth model, we confirm that the observed particle size development, including the DMD events, can be reproduced by non-volatile condensation, and thus without evaporation. In fact, when considering increasing contributions from a semivolatile compound, we find deteriorating agreement between the measurements and the model. Based on these results, it seems unlikely that evaporation of semivolatile compounds would





play a significant role in the DMD events at our measurement site. In the proposed non-volatile explanation, the DMD events are a result of the observed particles having spent an increasing fraction of their lifetime in a lower growth environment, mainly enabled by the lower precursor vapor concentrations further away from the measurement site combined with decreasing photochemical production of condensable vapors in the afternoon. The correct identification of the cause of

the DMD events is important as the fate and the climate-relevance of the newly formed particles heavily depends on it — if the particles evaporated, their net contribution towards larger and climatically active particle sizes would be greatly reduced. Our findings highlight the importance of considering transport-related effects in NPF event analysis, which is an often overlooked factor in such studies.

## 1.    Introduction

The largest uncertainty in our ability to quantify present-day climate change is related to the role of atmospheric aerosols (IPCC, 2021). The climate impact of aerosol particles mainly stems from the aerosol-cloud interactions. Thus, particles large enough (diameter $D_p$ > 50-100 nm) to act as cloud condensation nuclei (CCN) are of specific interest. In the atmosphere, the majority of CCN form via the growth of sub-CCN sized aerosol particles that originate from either atmospheric new particle formation (NPF), in which new particles are formed from vapor molecules, or emissions of small primary particles

(Merikanto et al., 2009;Kerminen et al., 2012;Paasonen et al., 2013;Gordon et al., 2017;Liu and Matsui, 2022). Therefore, the growth of small particles plays an essential role in aerosol-climate interactions, but the dynamics causing the particle growth and the chemical compounds involved are not yet adequately quantified (Zhang et al., 2012;Tröstl et al., 2016;Paasonen et al., 2018;Semeniuk and Dastoor, 2018).

In general, the most important driver of small particle growth up to CCN-sizes is considered to be the condensation of low-volatility oxidation products of volatile organic compounds (VOCs) (Smith et al., 2008;Laaksonen et al., 2008;Jimenez et al., 2009;Riipinen et al., 2012;Ehn et al., 2014;Kerminen et al., 2018;Dall'Osto et al., 2018). However, in many environments the measured concentrations of condensable species give underestimates for the growth of the newly formed particles (Tröstl et al., 2016;Qiao et al., 2021). There are several possible candidates for this mismatch, including undetected condensable

species and heterogeneous/particle phase formation of organic salts and oligomers, which reduce the volatilities of the sorbed compounds. Simultaneous measurements of the gas and particle phase compounds point towards lower-volatility products in the particle phase than expected directly from the gas-phase measurements (Baltensperger et al., 2005;Häkkinen et al., 2022;Zhao et al., 2022). These observations suggest that the hypothetically reversible condensation-evaporation dynamics governing aerosol growth might have a preferential direction towards accumulating more mass into the aerosols.

In light of the low, and lower-than-expected, volatilities of ambient aerosol particles, the observations of aerosol shrinkage events (or decreasing mode diameter (DMD) events, as referred to in this study) are somewhat perplexing. In these events,





the average diameter of a particle mode formed in an NPF event begins to decrease soon after the growth phase. At least at first glance, these observations suggest reduction in the particles' sizes due to evaporation, which would require a significant fraction of the particles' mass to consist of semivolatile species that can quickly partition in and out of the condensed phase. DMD events have been reported especially in subtropical (Yao et al., 2010;Backman et al., 2012;Cusack et al., 2013;Young

et al., 2013;Zhang et al., 2016;Alonso-Blanco et al., 2017), but also in temperate climate (Skrabalova, 2015;Salma et al., 2016). At many of these sites, DMD events are only observed during spring and summer months, which leads to speculations or conclusions stating that these events are enabled by aerosol evaporation in high temperatures. Dilution of vapor concentrations due to higher wind speeds or reduced photochemical production of condensable species are identified as other potentially determining factors. The evaporating species are thought to be semivolatile organic compounds (SVOCs), from

either anthropogenic or biogenic sources, or ammonium nitrate, which shows significant evaporation already at relatively low temperatures (Hong et al., 2017).

DMD events have, however, been observed in a wide range of temperatures and alternative explanations to evaporation exist. For example, Salma et al. (2016) report of a DMD event on a day with a median temperature of -2.2 °C with no

significant diurnal variation. In such conditions, Salma et al. (2016) deem temperature-driven particle evaporation an unrealistic explanation and propose changes in particle formation and growth rates during air mass transport as an explanation for the observation. Spatially and temporally differing growth rates are explicitly addressed in a modeling study by Kivekäs et al. (2016). In their study, Kivekäs et al. (2016) display an example case of a DMD event at Sammaltunturi, Finland, that can be explained by varying conditions during transport if the particles are set to grow slower during the night

and over the ocean. In this explanation, no reduction in the size of any individual particle is needed.

Understanding the dynamics leading to the observed DMD events is important for our capability to estimate the influence of NPF events on the climate: if the newly formed particles are reversibly evaporated back into the gas phase, their contribution towards CCN is practically nonexistent. Even partial evaporation would decrease the fluxes of particles above a certain size.

However, if the growth is essentially irreversible and the DMD results from transport effects, all of the observed particles can either potentially become CCN in the future or maintain their already-acquired status as a potential CCN.

In this study, we investigate the cause of DMD events in Hada Al Sham, Saudi Arabia, where such events were found to be exceptionally frequent (Hakala et al., 2019). We begin by looking into the local meteorological conditions and their changes

with respect to the occurrence of the DMD events. Our aim is to see whether there are clear and consistent changes in factors that could trigger particle evaporation at the measurement site. We then extend our analysis to account for moving air masses and investigate the potential source areas of the particles observed during the growth and DMD phases of the NPF events. In order to account for the varying conditions during air mass transport, we develop a Lagrangian single-particle growth model and evaluate the model-predicted diameter development of the NPF events against observations. We run the model in





multiple different configurations in order to cover a wide range of possible conditions and effects from poorly constrained or quantified factors. Our goal is to find out the main contributing factors to the DMD events observed in Hada al Sham with a specific focus on the question of whether particle evaporation is needed for explaining the observations or not. Similar methods to those developed and applied in this study could also be used on other sites to study the cause of DMD events.

## 2. Measurements and methods

Here, we shortly describe the measurements and data analysis on which this study is based on and refer the reader to Lihavainen et al. (2016) and Hakala et al. (2019) for more detailed description of the measurement site, instrumentation and data.

### 2.1 Measurement site and instrumentation

The measurements were conducted in Hada Al Sham (21.802 °N, 39.729 °E), a small city in western Saudi Arabia roughly 60 km east of Jeddah and the Red Sea, in February 2013 – February 2015. While there are no strong sources of anthropogenic emission in Hada Al Sham, the coastal area is a global hotspot of $SO_2$ emissions mainly due to the oil and gas industry in the region (Krotkov et al., 2016;Ukhov et al., 2020b). Biogenic sources of aerosol precursors in the surrounding areas are weak due to the arid desert climate (Sindelarova et al., 2014). The container, where the instruments were deployed, was situated at the Agricultural Research Station of King Abdulaziz University. The particle number size distribution (PNSD) in the mobility diameter range of 7–850 nm was measured with a twin DMPS (Differential Mobility Particle Sizer; Wiedensohler et al. (2012)) and meteorological parameters (temperature, relative humidity, wind speed and wind direction) with a Vaisala WXT weather station. In addition to the in situ data, we utilize the $SO_2$ Planetary Boundary Layer product (Li et al., 2013) derived from the Ozone Monitoring Instrument (OMI) satellite data (Levelt et al., 2006) for estimating the spatial variability of anthropogenic precursor vapors in the surroundings of Hada al Sham.

### 2.2 NPF event classification and progression

NPF event days were identified based on the criteria presented in Dal Maso et al. (2005). The NPF days were further separated into DMD (Decreasing Mode Diameter) days and non-DMD days, based on whether the mean diameter of the mode formed in an NPF event clearly started to decrease after the growth phase or not (see Fig. 1). In Alonso-Blanco et al. (2017), the DMD events (therein referred to as shrinkage events) were divided into three subclasses based on whether the DMD event was preceded by (1) NPF, (2) particle growth without NPF or (3) no NPF or growth. Although all of these types were found in our measurements, we will only focus on the DMD events preceded by NPF for two reasons. First, this was by far the most common type in our observations and second, we think the most meaningful way of finding explanation for the DMD events is to compare the conditions between "regular" NPF events and the NPF+DMD events, as well as the conditions during the growth phase to those during the DMD phase. In addition, our analysis of the NPF footprint areas



(Sect. 2.3.1) requires the start time of NPF to be defined, and this can only be determined for the DMD events that are observed to be preceded by NPF.

The progression of the NPF events is characterized by visually determining the following points in time from the PNSD data: (1) NPF is first observed in the smallest DMPS size-bins ($D_p = 7$ nm), (2) clear growth of the mode formed in the NPF event ends, (3) the mode diameter of the newly formed particles starts to decrease and (4) the decreasing mode diameter is no longer observed. These times are referred to as NPF start, Growth end, DMD start and DMD end, respectively (Fig. 1). The general criterion in defining the Growth end and DMD start times was to find the points in time when both the upper and lower edges of the NPF mode were seen to either increase or decrease simultaneously. On most of the DMD days, the Growth end and the DMD start times are the same. However, sometimes there is a transition period between the clear growth and DMD phases, and in some cases a growing mode and a DMD mode might also be observed simultaneously. In our analysis, the growth and DMD phases are defined as the periods between NPF start and Growth end, and DMD start and DMD end, respectively. Therefore, the possible transition periods (between Growth end and DMD start) are not included in either of these categories.

In Sect. 4.3, we simulate the NPF event progression using a Lagrangian single-particle growth model and compare the modeled particle diameters against observed ones. In this comparison, the observed diameters are defined using the output of an automated fitting algorithm (Hussein et al., 2005). The algorithm fits log-normal distributions to the measured PNSD data and gives the geometric mean diameters (GMDs) of the fitted distributions. From these, we select the GMDs representing the particle mode produced in an NPF event, apply a robust (outlier dampening) moving average filter and calculate the hourly geometric mean values. In Fig. 1, the selected GMDs from the fitting algorithm are shown with black dots and the average diameters, used in the model comparison, with white circles connected by a dashed line.




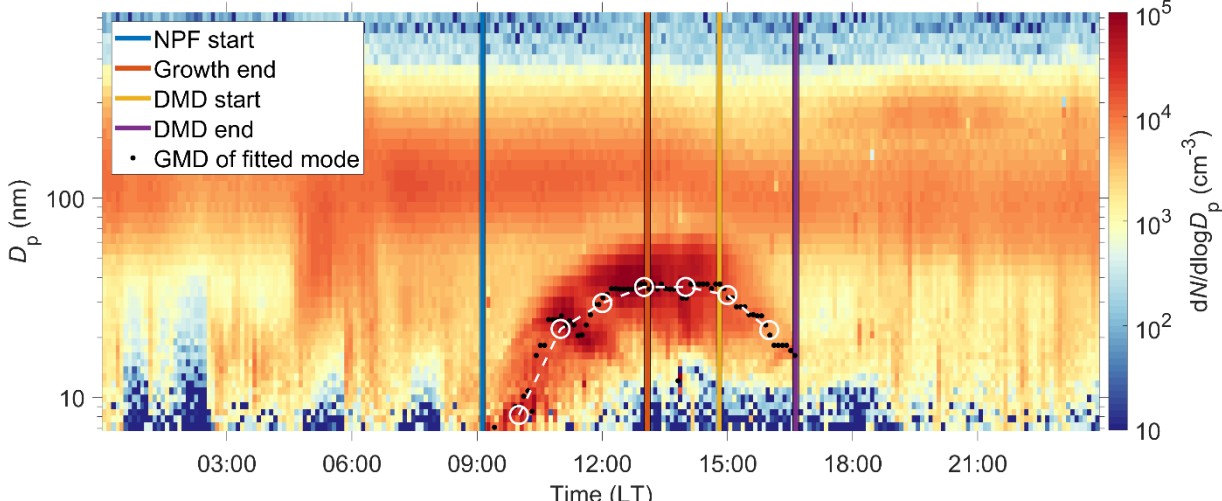

**Figure 1: Particle number size distribution measured by DMPS showing an NPF event with a decreasing mode mean diameter (after 15:00 LT; UTC+3) in Hada Al Sham on July 9, 2013. The figure also illustrates an example of the times describing the progression of the NPF events (colored vertical lines), the geometric mean diameters (GMDs) for the NPF-related particle mode from the fitting algorithm (black dots) as well as the hourly average diameters used in the model comparison (white circles).**

## 2.3 Air mass history

We use a Lagrangian particle dispersion model FLEXPART v9.02 (Pisso et al., 2019) with ECMWF (European Centre for Medium-Range Weather Forecast) operational forecast data (0.15° horizontal and 1 h temporal resolution) to compute hourly air mass retroplumes for the study period. In each simulation, 50 000 passive air tracers are released from 0-100 m above the location of the measurement site and traced backwards for 3 days. The gridded residence times of the tracer retroplumes are used for estimating the source areas of the NPF-related particles while the plume trajectories (retroplume centroids) are used in the Lagrangian single-particle growth model (Stohl et al., 2002;Seibert and Frank, 2004).

### 2.3.1 NPF footprint area

In Sect. 4.2, we utilize the air mass retroplumes to estimate the potential source areas of NPF-related particles. This is done by calculating the locations of air masses, observed at different stages of the NPF progression, at the onset time of the regional NPF event. We refer to this potential source area as the NPF footprint area. In terms of the FLEXPART output, the NPF footprint area for particles observed at $t_{obs}$ is defined as the residence time of the air mass tracers, released at $t_{obs}$, in the lowest 500 m (a.g.l.) at the backward calculation time of $t_{obs}$ – NPF start. Here NPF start is as defined in Sect. 2.2 but rounded down to the nearest full hour. The rounding-down is done to compensate for the determination of NPF start at $D_p = 7$ nm, while the actual NPF takes place earlier in smaller particle sizes. Note that interpreting the NPF footprint area as the area where new particle formation took place includes the assumption that regional NPF occurs simultaneously in the whole



study domain. However, even if this is not the case, the NPF footprint area is still meaningful in showing the air mass location at the time when NPF was observed to take place around the measurement site.

## 2.4 Lagrangian single-particle growth model

In our model, the idea is to represent particle (formation and) growth within a system consisting of a non-volatile compound and some potentially volatile compound, whose evaporation could cause the observed DMD events. The compounds are referred to with the subscripts non and vol, respectively, and the *potentially volatile* compound will be referred to as the volatile compound for short. The compounds are also interchangeably referred to as the non-volatile/volatile "species" or "components" depending slightly on the context. In practise, the non-volatile compound represents sulfuric acid neutralized

with base molecules, and the volatile compound a low- to semivolatile oxidation product of some anthropogenic organic precursor. Such vapors would produce particles consisting mainly of ammonium sulfate and organic matter, which are reported to be the main components of fine particle mass in the United Arab Emirates (Kesti et al., 2022), representing a similar region to that of our measurements. The relevant values for these compounds are presented in Table 1. Note that in terms of initial particle formation, the non-volatile compound in practice represents sulfuric acid neutralized with an

abundant base stronger than ammonia, e.g. dimethyl amine, which can form effectively stable heterodimers with sulfuric acid and allow for kinetically controlled particle formation without a free energy barrier (Cai et al., 2022). Although the non-volatile assumption is made here mostly due to the lack of measurement data and to simplify the results interpretation, conditions close to such might also prevail in the real atmosphere (Kürten et al., 2018;Cai et al., 2022).

**Table 1. Properties of the considered compounds in the single-particle growth model. The saturation vapor pressure range of $P_{sat,vol}(T_0)$ = 1e-9…1e-5 Pa corresponds to a saturation mass concentration range of $\log_{10}(C_{sat})$ = -2.8…1.2 μg m$^{-3}$ at T = 300 K using $\Delta h_k$ = 80 kJ mol$^{-1}$. This covers the transition between low-volatility and semivolatile organic compounds (at $\log_{10}(C_{sat})$ = -0.5) as defined in Donahue et al. (2012).**

|  | Mass $m_k$ (amu) | Density $\rho_k$ (=$\rho_p$) (kg m$^{-3}$) | Saturation vapor pressure $P_{sat,k}(T_0$ = 278 K) (Pa) | Surface tension $\sigma_k$ (N m$^{-1}$) | Enthalpy of vaporization $\Delta h_k$ (kJ mol$^{-1}$) |
|---|---|---|---|---|---|
| non-volatile | 145 | 1500 | 0 | - | - |
| volatile | 300 | 1500 | 0, 1e-9…1e-5, ∞ | 0.03 | 40, 60, 80, 100, 120 |



### 2.4.1 Formulation of particle growth rate

An expression for the growth rate of a spherical particle as a function of its mass change rate can be obtained by differentiating the mass expressed as a product of density and volume (Seinfeld and Pandis, 2016):

$$\frac{dm_\mathrm{p}}{dt} = \frac{d(\rho_\mathrm{p} V_\mathrm{p})}{dt} = V_\mathrm{p}\frac{d\rho_\mathrm{p}}{dt} + \frac{1}{2}\pi\rho_\mathrm{p}D_\mathrm{p}^2\frac{dD_\mathrm{p}}{dt} \approx \frac{1}{2}\pi\rho_\mathrm{p}D_\mathrm{p}^2\mathrm{GR}$$

$$\mathrm{GR} = \frac{2}{\pi\rho_\mathrm{p}D_\mathrm{p}^2}\frac{dm_\mathrm{p}}{dt}\ . \tag{1}$$

Here the subscript p refers to particle, $m_\mathrm{p}$ (kg), $\rho_\mathrm{p}$ (kg m$^{-3}$) and $D_\mathrm{p}$ (m) are the particle mass, density and diameter, respectively, and GR (m s$^{-1}$) is the growth rate (time derivative of particle diameter). In derivation of Eq. (1) changes in

particle density were assumed to be minor.

Changes in particle mass occur by the collision and coalescence of molecular species or smaller particles to the inspected particle, or by the dissociation of material from the particle. In our single-particle growth model, we will not consider particle-particle collisions and thus we formulate the GR as the condensational growth rate (GR$_\mathrm{cond}$) using the condensation

and evaporation rates of molecular species k (namely either non or vol):

$$\mathrm{GR}_{\mathrm{cond},k} = \frac{2}{\pi\rho_\mathrm{p}D_\mathrm{p}^2}\sum_k(K_{k,p}C_k - \gamma_k)m_k\,, \tag{2}$$

where $K_{k,p}$ (m$^3$ s$^{-1}$) is the collision coefficient between the particle and species k, $C_k$ (m$^{-3}$), $\gamma_k$ (s$^{-1}$) and $m_k$ (kg) are the number

concentration, evaporation rate and the mass of species k, respectively. The evaporation rate $\gamma_k$ can be written as:

$$\gamma_k = K_{k,p}\frac{P_{\mathrm{sat},k}(T)}{k_\mathrm{b}T}x_k\,\Gamma_{\mathrm{a},k}\exp\left(\frac{4\sigma_k m_k}{k_\mathrm{B}T\rho_k D_\mathrm{p}}\right)\,, \tag{3}$$

where $K_{k,p}$ is again the collision coefficient, $P_{\mathrm{sat},k}(T)$ is the temperature dependent saturation vapor pressure of pure k over a

flat surface (Pa; kg m$^{-1}$ s$^{-2}$), which is converted into vapor concentration (m$^{-3}$) by dividing with the Boltzmann constant $k_\mathrm{B}$ (m$^2$ kg s$^{-2}$ K$^{-1}$) and temperature (K). $x_k$ is the molar fraction, $\Gamma_{\mathrm{a},k}$ is the activity coefficient and $\sigma_k$ is the surface tension (N m$^{-1}$; kg s$^{-2}$). The multiplication by the molar fraction in Eq. (3) describes the reduction in the saturation vapor pressure of k in a (dilute) solution, according to the Raoult's law. The saturation vapor pressure (or evaporation rate) can be further decreased or increased depending on the relative strength of interactions between k and the solvent molecules as compared to those in

pure k. This interaction strength is described by the activity coefficient $\Gamma_{\mathrm{a},k}$. In our calculations, however, $\Gamma_{\mathrm{a},k}$ is set to 1. The





last term in Eq. (3), known as the Kelvin term, describes the increasing tendency to evaporate with increasing surface curvature (decreasing particle size) due to reduced number of neighboring molecules. The temperature dependency of the saturation vapor pressure is expressed with the Clausius-Clapeyron equation:

$$P_{\text{sat,k}}(T) = P_{\text{sat,k}}(T_0) \exp\left(\frac{\Delta h_k}{R}\left(\frac{1}{T_0} - \frac{1}{T}\right)\right),$$ (4)

where $P_{\text{sat,k}}(T_0)$ is the saturation vapor pressure of pure k over a flat surface at a reference temperature $T_0$. $\Delta h_k$ is the enthalpy of vaporization (J mol$^{-1}$), which has been assumed to be constant as a function of temperature, and $R$ is the molar gas constant (J mol$^{-1}$ K$^{-1}$).

In our growth model, we will only consider collisions between vapor molecules and particles in (or near) the ultrafine range ($D_p < 100$ nm). Thus, the collision coefficient $K_{k,p}$ in Eqs. (2) and (3) is approximated using the free molecular regime collision coefficient for hard spheres (with no interactions over distance):

$$K_{k,p} = \frac{\pi}{4}\left(D_p + D_k\right)^2\left(\bar{c}_p^{\ 2} + \bar{c}_k^{\ 2}\right)^{\frac{1}{2}},$$ (5)

where $D_p$ and $D_k$ are the diameters of the particle and vapor molecule, respectively, and $\bar{c}_p$ and $\bar{c}_k$ are their mean thermal velocities:

$$\bar{c}_{p,k} = \sqrt{\frac{8k_B T}{\pi m_{p,k}}}.$$ (6)

### 2.4.2 Estimating the concentrations of the condensing species

In the Lagrangian framework, the concentrations of the condensing species need to be evaluated along the air mass transport routes. Essentially, this would require information on the spatial and temporal variability of the species within the whole study domain. As no such measurement network exists, we utilize the satellite-derived [SO$_2$] field to describe the spatial variability of the precursor vapor concentrations and use a simple pseudo-steady-state proxy equation, similar to those in Petäjä et al. (2009);Mikkonen et al. (2011);Kontkanen et al. (2016), to estimate the concentrations of the condensing species. The same basic formulation, whose assumptions and details are discussed further in the following paragraphs, is used for calculating the spatiotemporally varying concentrations of both the non-volatile ($C_{\text{non}}$) and volatile ($C_{\text{vol}}$) compounds:





$$C_{ij}(t) = k \frac{[SO_2]_{ij}\,rad_{ij}(t)\,BLH_{ij}^{eBLH}(t)\,WS(t)^{eWS}}{CS(t)^{eCS}}, \qquad eBLH, eWS, eCS \in [-1, 1] \tag{7}$$

$$C_{non} = f_{non}C_{ij}(t), \quad f_{non} \in [0, 1] \tag{8}$$

$$C_{vol} = f_{vol}C_{ij}(t), \quad f_{vol} \in [0.2, 5], \tag{9}$$

Here $[SO_2]$ is the average daytime concentration of $SO_2$ (DU; Dobson units, expressing the atmospheric column burden) for NPF event days from the satellite retrievals (shown in Fig. 6d). The subscript ij represents variable dependency on the latitude and longitude coordinates of a trajectory. rad is the calculated theoretical no-sky radiation (W m$^{-2}$), BLH is the boundary layer height (m) obtained from the ECMWF meteorological data used in the trajectory calculations (based on the bulk Richardson method) and WS and CS are the wind speed and condensation sink (describing the loss rate of the condensing species to preexisting particles) measured at Hada al Sham. $k$ is a scaling factor, whose value and units depend on the exponents eBLH, eWS and eCS. The value of $k$ is chosen so that the growth from the non-volatile component alone at $C_{non} = C_{ij}(t)$ (i.e. $f_{non} = 1$ in Eq. (8)) exceeds the typically observed growth rates, and the units of $k$ convert the units of the final concentration to m$^{-3}$. We therefore use the observed PNSD development (growth rate) to constrain the maximum concentration of the non-volatile compound in the absence of gas-phase measurements. The concentrations and relative contributions of both the non-volatile and volatile components are then varied in different model runs by varying the concentration multiplying factors $f_{non}$ and $f_{vol}$ in Eqs. (8) and (9). In case of the volatile component, we additionally vary the saturation vapor pressure and the enthalpy of vaporization (as shown in Table 1) in order to cover the relevant range of condensation-evaporation dynamics.

$SO_2$ is the precursor for the non-volatile component (neutralized sulfuric acid in a system assumed to be saturated with base molecules) and it is mainly emitted from anthropogenic sources. As the volatile (organic) compound is also likely to be of anthropogenic origin, we assume in Eq. (9) that the precursors for the volatile compound have the same spatial distribution as the non-volatile one, described by the $[SO_2]_{ij}$. The multiplication of $[SO_2]$ and rad represents the photochemical production of condensable vapors via OH oxidation, which is likely to be the dominant formation pathway for both sulfuric acid and condensable products from anthropogenic VOCs during daytime (Bourtsoukidis et al., 2019;Srivastava et al., 2022). We do not, however, expect the concentrations of the condensing species go to zero with the radiation, and therefore set a minimum value of 100 W m$^{-2}$ for the radiation in the calculations after noon, which represents maintained production of the condensable compounds via other oxidation pathways. This roughly corresponds to observations of sulfuric acid concentrations being one order of magnitude lower (compared to the noon) during non-solar hours, when the production occurs via stabilized Criegee intermediates from the ozonolysis of alkenes (Dada et al., 2020). The same relative strength of



the production pathways between the solar and non-solar hours is also assumed for the volatile organic compound, as roughly observed e.g. in south-eastern US (Krechmer et al., 2015) in a low $NO_x$ and high radiation environment.

The [$SO_2$] field implemented in the calculations does not vary in time and only represents the total atmospheric column burden of $SO_2$ instead of the surface (or boundary layer) concentration, which are presumably more relevant for our calculations. Because of this, the BLH and WS terms are included to describe the possible variation in the precursor vapor concentrations caused by differing dilution conditions. CS describes the loss of the condensing vapors, which is assumed to be in a pseudo-steady-state with the production. The strengths of the aforementioned effects are determined by the exponents eBLH, eWS and eCS, that are considered unknown and varied in different model runs. Even though the dilution by BLH and

WS would only include negative values for eBLH and eWS in Eq. (7), we extend the inspected range to also include positive exponents. Similarly, we extend the range of eCS into negative exponents, since in Hakala et al. (2019) we found the particle formation and growth rates to be positively correlated with CS, which is likely caused by shared sources of precursor vapors and particles acting as the sink. Varying signs and magnitudes for the CS term exponent have been found for similar proxy equations in different environments (Mikkonen et al., 2011). Further regarding the CS term, we wish to point out that using

the CS measured at our station is not ideal in the Lagrangian framework, as we do not expect the sink to be horizontally homogeneous. We also note that in our calculations, the CS term affects similarly both the $C_{non}$ and $C_{vol}$, regardless of the actual saturation ratio of the volatile component.

In addition to the effects explicitly shown and discussed here, we experimented with several other factors that could affect

the concentrations of the condensable species experienced by the air mass. These included testing varying exponents for the [$SO_2$] and rad terms, using temporally varying [$SO_2$] fields and applying a concentration-dependency based on the height of the trajectory with respect to the boundary layer. We also considered a case where the boundary layer would only affect the concentrations during its growth but not during its collapse. However, none of these were found to improve our results, albeit not all the possible combinations of effects were tested.

**2.4.3 Modeled particle diameter and the contributions from condensational growth and transport**

In our model, the diameter development of particles arriving at the measurement site at different observation times ($t_{obs}$) are modeled separately. For a particle, observed at $t_{obs}$, the diameter at time $t$ during its transport is calculated as

$$D_{\text{p,model}}(t_{\text{obs}}, t \in [0, t_{\text{obs}}]) = D_{\text{p,init}} + \int_{t=0}^{t} \text{GR}_{\text{cond,non}}\left(f_{\text{non}}C_{\text{ij}}(t)\right) + \text{GR}_{\text{cond,vol}}\left(f_{\text{vol}}C_{\text{ij}}(t)\right) \, \mathrm{d}t \,, \qquad (10)$$



where $GR_{cond}$ is the growth/evaporation rate given by Eq. (2), $D_{p,init}$ is the diameter of the initial "particle" (diameter of the non-volatile vapor molecule, obtained from its mass and density) and $C_{ij}(t)$ is the baseline concentration of condensing species (given by Eq. (7)) at the location where a trajectory arriving at the measurement site at $t_{obs}$ is located at time $t$. Here, new particle formation occurs barrierlessly by the condensation of the non-volatile component onto the initial "particle".

With larger particle diameters (reduced Kelvin effect), also the volatile compound can contribute to the growth depending on its saturation ratio. Each day is modeled separately and $t = 0$ refers to the $0^{th}$ hour of the day. The integration is performed with a 6 min time step to better account for particle diameter and composition-dependent changes, while the particle location and the input data has a 1 hour resolution.

The contributions from condensational growth or evaporation ($GR_{true}$) and transport ($GR_{transport}$) to the modeled diameter changes between subsequent "observation" times are separated, and defined as

$$GR_{true}(t_{obs}) = \frac{D_{p,model}(t_{obs}, t_{obs}) - D_{p,model}(t_{obs}, t_{obs} - \Delta t)}{\Delta t} \tag{11}$$

$$GR_{transport}(t_{obs}) = \frac{D_{p,model}(t_{obs}, t_{obs} - \Delta t) - D_{p,model}(t_{obs} - \Delta t, t_{obs} - \Delta t)}{\Delta t}. \tag{12}$$

Hence, $GR_{true}$ is the diameter change rate of the particle observed at $t_{obs}$, before its arrival at the measurement site, while $GR_{transport}$ is the difference in the sizes of consecutively arriving particles (particles observed at $t_{obs}$ and $t_{obs}$- $\Delta t$) at the arrival time of the first particle (at time $t_{obs}$-$\Delta t$), divided by the time step. That is, if the later-observed particle was already

larger/smaller than the first-observed particle at the arrival time of the first particle, this difference is attributed to transport, as it originates from the different transport paths taken by the particles. With these definitions, the apparent growth rate in the modeled observations (referred simply to as $GR_{model}$), which conceptually corresponds to the apparent growth rate directly obtainable from the observations, is

$$GR_{model}(t_{obs}) = GR_{true}(t_{obs}) + GR_{transport}(t_{obs})$$

$$GR_{model}(t_{obs}) = \frac{D_{p,model}(t_{obs}, t_{obs}) - D_{p,model}(t_{obs} - \Delta t, t_{obs} - \Delta t)}{\Delta t} \tag{13}$$

as it should be. Note that the common practice of referring to the directly observed diameter changes as *growth rates* always includes the assumption that the contributions from transport (and coagulation) are zero or negligible, given that the growth

rate is understood as the real diameter change that an individual particle would experience. Here, we use the term *apparent growth rate* to refer to the observed diameter changes that can include contributions from both the real growth/shrinkage





immediately prior to the observations, as well as the earlier variations in the growth rate caused by differing environmental conditions along the different transport paths.

### 2.4.4 Model evaluation metrics

The model performance is evaluated against observations using three different metrics. The three evaluation metrics are:

Evaluation metric 1: correlation between the logarithms of the hourly observed and modeled particle diameters,

Evaluation metric 2: goodness of fit between observed and modeled diameter changes with a line passing through the origin during well-defined growth and DMD periods and

Evaluation metric 3: exponential of the mean absolute log error between the hourly observed and modeled diameters, i.e.

$$\text{Evaluation metric 3} = \exp\left(\overline{\left|\log(D_{\mathrm{p,model}}) - \log(D_{\mathrm{p,obs}})\right|}\right). \tag{14}$$

The first two evaluation metrics only depend on how well the observed shape of the diameter development is reproduced by the model, irrespective of the actual numerical agreement. The second evaluation metric is specifically included to indicate if the model is able to produce both increasing and decreasing diameters, as the correlation with a line passing through the origin should not be strong in case of monotonic growth, while the diameter correlation (metric 1) could still give relatively high values. The third evaluation metric gives the average value with which the model $D_{\mathrm{p}}$ needs to be multiplied or divided with in order to arrive at the observed value, and therefore assigns same error for modeled diameters that are a factor of $x$ larger or smaller than the observed ones. This metric depends on the numerical agreement between the observed and modeled diameters and therefore also constrains the concentrations of the condensing species.

### 3. Hypotheses based on our previous results

In this section, we will briefly summarize some of the relevant results of our previous study, which focused on the general characteristics of NPF at the Hada al Sham site (Hakala et al., 2019), in order to lay the foundation for the work presented in this study. We will also hypothesize the possible causes of the DMD events based on the previously obtained results. Here, the additional motive is to explain and illustrate the conceptually more complex process resulting in DMD, which we refer to as the apparent shrinkage process.

NPF events were observed on 73 % of the measurement days (n = 454) and 76 % of these NPF events were further classified as DMD events (Hakala et al., 2019). The frequent occurrence of NPF was found to be linked to the transport of anthropogenic emissions from the coastal regions, caused by a highly regular sea and land breeze circulation. The infrequent presence of clouds in the region (Stubenrauch et al., 2010) is also favorable for NPF. Clear nonevent days were only observed when the sea breeze was blocked by strong easterly winds from the inland, preventing the transport of NPF





precursors to the measurement site. NPF events were found to start (observed at $D_p$ = 7 nm) a few hours after the sunrise throughout the year, and the DMD phase typically started in the afternoon around 6 h after the NPF start. A few hours later, the particles clearly associated with NPF typically disappeared completely from our observations, regardless of whether a clear DMD phase was seen or not. The disappearance of the NPF-related particle mode suggests that NPF is only taking

place on a spatially limited area (Hussein et al., 2009). Especially in the DMD cases, the mode disappearance can also be related to the smaller particles being lost to coagulation as a result of increased coagulation efficiency with decreasing particle size. The mean lifetime of e.g. a 10 nm particle is in the order of a few hours with the typical coagulation sink at the measurement site ($10^{-4}$ s$^{-1}$). While there was no clear seasonal variation in the overall NPF frequency, the DMD events were found to be more common during the warmer and windier summer months (Hakala et al., 2019).

Based on the initial result obtained in our previous study, we briefly speculated the reasons that could be causing the observed DMD events in Hada Al Sham. The first possibility is particle evaporation due to decreasing vapor concentrations and increasing volatility in the afternoon. The decreasing vapor concentrations could possibly be caused by decreasing emissions, increasing sink or by increasing dilution with increasing wind and BLH in the afternoon. The volatility changes

could be related to the changes in temperature. The onset of the DMD events was typically found around the daily maximum temperature, BLH and wind speed (Hakala et al., 2019). Almost all of the DMD events started after the time of maximum solar radiation, when the concentrations of condensable vapors may also decrease due to their decreasing photochemical production rate.

Another possible explanation for the DMD events is a process which we refer to as apparent shrinkage (illustrated in Fig. 2), and in which no real shrinkage of the particles occurs. Apparent shrinkage can be caused by consecutive observations of particles that have grown less during their lifetime than the previously observed particles (Kivekäs et al., 2016). The possibility of this process is supported by the observation that the NPF events seemed to occur only on a limited area, whose extent roughly matched that of a high SO$_2$ concentration area determined from satellite observations (Hakala et al., 2019).

Particle formation and growth rates associated with the NPF are expected to decrease, when moving further away from the high concentration area. Thus, particles formed further away from the measurement site are likely to be smaller (at a given time) than those formed in the high concentration area. Since, on average, we observed no change in the wind direction during the transition from the growth phase to the DMD phase, it is likely that the air masses observed during the DMD phase would have travelled over the same high concentration areas as the previous air masses. Therefore, in addition to the

slower initial growth, the occurrence of the apparent shrinkage requires that the particles observed during the DMD phase also grow less during the transport over the high concentration areas; otherwise, they would grow at least as much as the previous particles, due to the longer growth time resulting from the later time of observation. The lesser growth in the high emission area can occur if the particles travel faster over the high concentration area or if the particle growth rates decrease as the day progresses. The decreasing growth rate can be caused by all the same phenomena as discussed above in the





context of particle evaporation. However, in the case of particle evaporation, the effects of these phenomena would need to be so strong that not only would the growth rates decrease but turn negative. We note that in the initially clean and remote air masses, new particles could also form later in the day (compared to those initially in the high concentration area) once they are transported over the strong emission sources. Such delayed NPF start could also result in a DMD event, especially since

in this case the total growth time between the formation and the detection of the particles would no longer increase as the day progresses, if constant wind speed is assumed.

In summary, for the apparent shrinkage both conditions (1) and (2) below must be fulfilled with any combination of (a) and/or (b):

**1)** Air mass, in which DMD is observed, must be outside the high concentration area during the regional NPF event start. This leads to:

a) slower initial growth due to lower vapor concentrations or

b) later formation of particles once the air mass is transported to an area with sufficiently high vapor concentrations for NPF.

Importantly in both (a) and (b), the result is smaller (or no) particles in the DMD air mass compared to those initially inside

the high concentration area at any time $t$.

**2)** Particles in the air mass described in (1) must grow less during their transport over the high concentration area compared to the particles initially in the high concentration area

This can be achieved by:

a) decreased concentration of condensable vapors in the high concentration area (possibly caused by enhanced dilution due to increasing BLH and wind speed, reduced production due to weakening photochemistry, increased volatility due to increased temperature, decreased emissions or increased sink)

b) faster transport across the high concentration area (in this case slower growth in the high concentration area is not necessary).

This will allow the particles to be observed at a smaller size compared to the previous ones, provided that the initial diameter setback in (1) is larger than the additional diameter gained due to the later observation time.





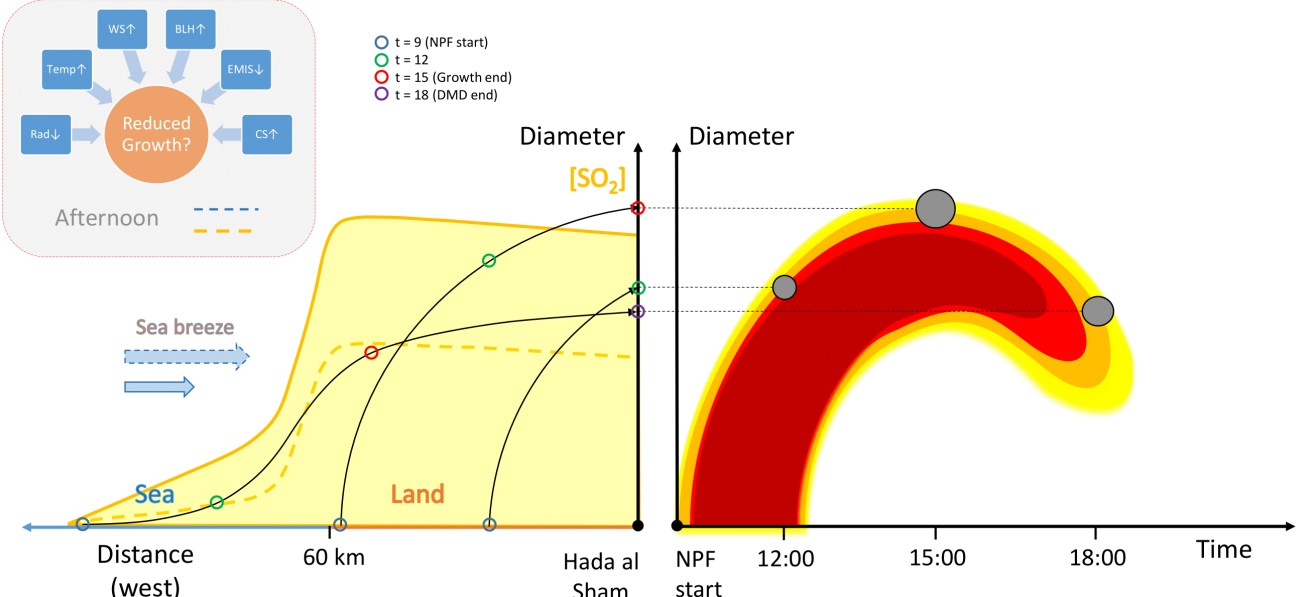

**Figure 2: A schematic figure illustrating the apparent shrinkage process and the factors possibly contributing to its occurrence. Left panel: longitudinal cross section of the measurement site (located at the origin) and its surroundings with the horizontal axis representing distance towards west. The yellow shaded area represents the average daytime spatial distribution of the SO₂**
**concentration, which is high over land but drops significantly over the ocean (see Fig. 6d). The black arrows represent particles formed in a regional NPF event at different locations and the vertical axis corresponds to their diameter as they are transported towards the measurement site by the sea breeze. The particle diameters and horizontal locations at different times of day are specified with the colored circles. Once the particles arrive at the measurement site, their diameters are plotted on the right panel as a function of time to illustrate the observed particle size distribution. In the afternoon, several factors can contribute to reduced**
**growth of particles (see text and box in the upper left corner), which could result in the observation of a decreasing mode diameter even if the individual particles are constantly growing in size.**

## 4. Results

### 4.1 Meteorological conditions during growth and DMD periods

If local meteorological conditions and their changes are triggering particle evaporation and causing the DMD events, we
expect to find significant differences in these conditions between the DMD and non-DMD cases. We will first compare the
average meteorological conditions in the afternoon (12:00-18:00) over the annual cycle between days when NPF events were
observed either with or without a DMD phase (Fig. 3; DMD and non-DMD days, respectively). Many of the previous studies
on DMD events suggest temperature and wind speed to play major roles in the occurrence of the DMD events (Alonso-
Blanco et al., 2017). The significance of the differences between the DMD and non-DMD days in Fig.3 is evaluated using
the Mann–Whitney U-test (at 5% significance level) separately for each month. Statistically significant differences are
highlighted with green shading if they support the speculated causes that could be triggering particle evaporation, i.e. if
temperature, wind speed and boundary layer height are higher or if the radiation is lower for the DMD days. Statistically
significant differences towards the opposing direction are highlighted with red.



Looking first at the temperatures, we find mostly similar conditions between the DMD and non-DMD days, and the only significant difference in February shows higher temperatures on the non-DMD days (Fig. 3a). We can also see that the DMD events can occur even on the lowest temperature days during the winter. This indicates that the afternoon average temperatures are not a controlling factor in the DMD occurrence in Hada al Sham. However, we note that even the wintertime temperatures at this site are high compared with many other locations.

In the case of the daytime wind speeds (Fig. 3b), more significant differences are found between the DMD and non-DMD days. The wind speeds are found to be higher on the DMD days on 6 of the 12 months, mostly during the winter. During the generally windier summer months (month nos. 4-9), the number of non-DMD days is overall low, with a maximum of four cases in each month. Examining the fraction of the non-DMD events as a function of wind speed shows some interesting statistics: when the daytime average wind speed exceeds 5.5 m s$^{-1}$, only ~7 % (7 out of 97) of the NPF days were non-DMD days, whereas with WS < 3.5 m s$^{-1}$ the fraction is ~70 % (24 out of 35). Therefore, high wind speeds seem clearly favorable for the DMD events, and the seasonal changes in wind speed could possibly explain the higher frequency of the DMD events in summer.

In Fig. 3c and d, we show the boundary layer height (from the ECMWF meteorological data) and the theoretical no-sky radiation. The BLHs are mostly similar and the significant differences in February and August show higher BLHs on non-DMD days, which does not support stronger (vertical) dilution as a cause of the DMD events. No difference is either seen or expected in the theoretical radiation conditions and the values are shown here to illustrate the overall seasonal variability.





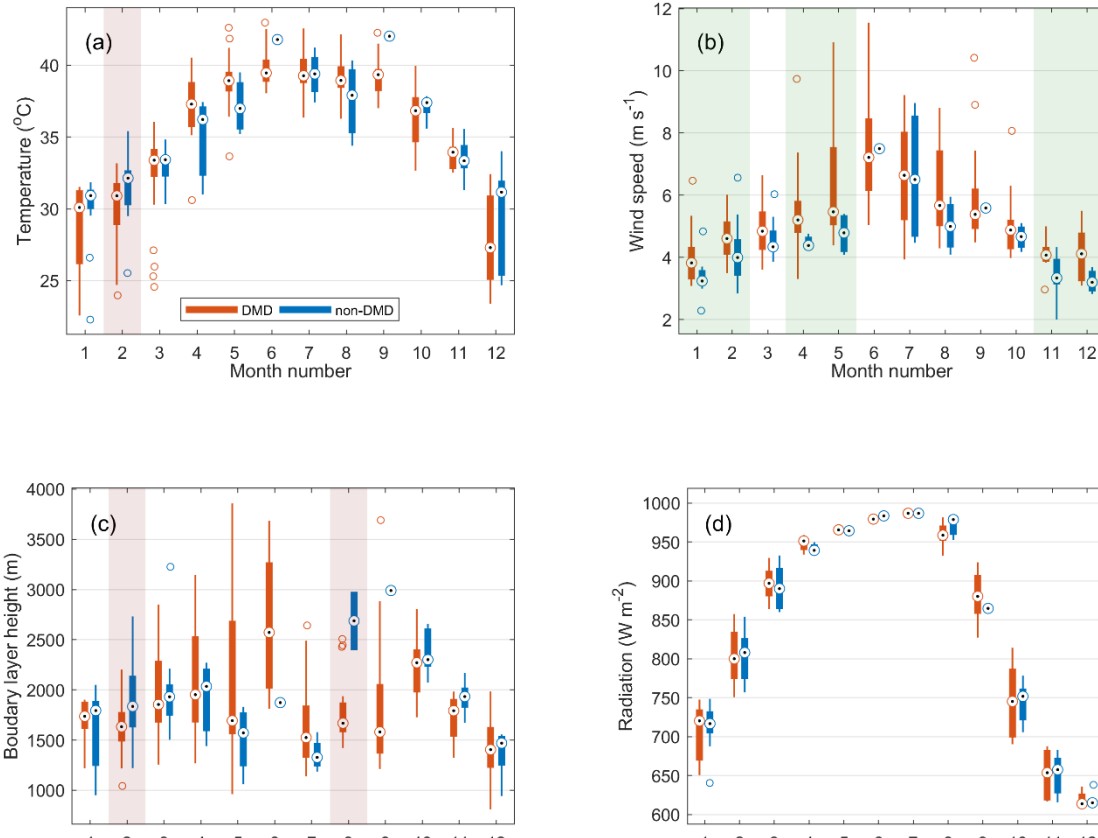

**Figure 3: Monthly distribution of afternoon (12:00-18:00 LT) mean (a) temperature, (b) wind speed, (c) boundary layer height and (d) theoretical no-sky radiation, separately for DMD days (red) and non-DMD days (blue). The time window is selected to cover the typical onset times of the DMD events (Hakala et al., 2019). Statistically significant differences (one-sided Mann-Whitney U-test at 5% significance level) between the DMD and non-DMD cases are highlighted with green shading if the difference is in line with the hypothesized triggers for particle evaporation and with red shading for significant differences towards the opposite direction. For each box, the central mark indicates the median, and the bottom and top edges of the box indicate the 25th and 75th percentiles, respectively. The whiskers extend to the most extreme data points not considered outliers, and the outliers (distance from the top or bottom edge of the box more than 1.5 times the interquartile range) are plotted individually using the 'o' symbols.**

In Figs. 4 and 5, we focus on the diurnal variation in the metrological conditions (temperature, wind speed, boundary layer height and the theoretical no-sky radiation) to obtain a closer look on the potential causes of the DMD events. Similar figures for relative humidity and wind direction are provided in the Appendix (Figs. A1 and A2). In these figures, we do not separate the days into DMD and non-DMD days but instead consider all NPF events and make the separation only based on the current phase (growth/DMD) of the event (See Sect. 2.2 and Fig. 1 for details). In Fig. 4, we compare the conditions of NPF



events in the DMD phase with those that still display growth at the same time of day (on another day). In Fig. 5 we focus on the changes in the meteorological conditions specifically at the onset times of the DMD events and compare them against the changes on days when growth still continues at the same time. This investigation should more specifically pinpoint any significant differences between the growth and DMD cases. In both figures, the results are shown separately for summer and
winter in order to reduce the bias that would arise from the DMD events being more frequent during summer. The significance of the differences is evaluated in the same manner as in Fig. 3.

The main findings from Figs. 4 and 5 are summarized below:

Temperature is generally not higher during the DMD phase, and in winter it often seems to be even lower during the DMD
phase than during the growth phase (Fig. 4a, b). The observed differences in winter should not be due to an uneven representation of different months in the DMD and non-DMD cases since the DMD events are biased towards the warmer months, as confirmed by the higher theoretical radiation on the DMD days (Fig. 4g, h). Changes in temperature during the DMD transition can be either positive or negative, depending mostly on the time of day, and the changes are neither large nor significantly different from those during the growth hours (Fig. 5a, b). The two cases of significant differences (at 16 LT
summer and 15 LT winter) show temperatures decreasing more around the DMD start than during growth at the same time. None of these findings support temperature-driven evaporation as the cause of the DMD events.

Wind speeds are consistently higher during the DMD periods, although in winter slightly less so (Fig. 4c, d). WS is typically increasing around the DMD transition in summer but in winter this is less clear (Fig. 5c, d). The change is typically not
significantly higher around the DMD start than during the growth hours. The higher wind speed during DMD could support dilution-driven evaporation or DMD by transport of particles from further-away regions, as the higher wind speeds leave less time for particles in the DMD air masses to grow as they travel over the high concentration areas. Wind direction is typically from the east during the day but the DMD phase shows some preference towards north-easterly winds while south-easterly winds are more common during the growth hours (Fig. A1c, d). No clear differences are seen in the change of wind direction
around the DMD onset times (Fig. A2c, d).

BLH is generally lower during the DMD phase, especially in summer (Fig. 4e, f). The DMD start times can be associated with both increasing and decreasing BLH (Fig. 5e, f). The significant differences show smaller increases/larger decreases in the BLH during the DMD transition. These results do not support DMD being caused by increased vertical mixing.

In the case of the theoretical no-sky radiation (Figs. 4g, h and 5g, h) any significant differences are a result of uneven distribution of the cases between the different months. The theoretical radiation values are shown to display that radiation is almost always decreasing during the DMD phase and thus the reduced photochemical production of condensable vapors could contribute to the DMD events. However, decreasing radiation at some point during the NPF progression happens with





most regional NPF events around the world and this does not systematically trigger DMD events. Also here, the growth phase often continues even after the radiation starts to decrease and some of the DMD events are found to start while the radiation is still increasing.

5    Overall, the only consistent difference in the meteorological conditions is the higher wind speeds on the DMD days and during the DMD phase. However, since (1) the magnitude of the difference is quite small compared to the growth cases, (2) the DMD onset is not related to any unusually large changes in WS, (3) the wind direction does not change significantly during the transition and (4) the BLH is not consistently higher or increasing significantly, we consider it unlikely that a wind-driven increase in dilution would be enough to trigger significant particle evaporation. The role of evaporation in the

10    DMD events is still examined further in Sect. 4.3 using the Lagrangian single-particle growth model. In the next section, however, we will inspect the prerequisites for the transport-driven apparent shrinkage process.





**Figure 4: Comparison of hourly mean (a, b) temperature, (c, d) wind speed, (e, f) boundary layer height and (g, h) theoretical no-sky radiation between NPF events in growth phase (blue bars) and DMD phase (orange bars) separately for summer (months: Apr–Sep, left column) and winter (months: Oct–Mar, right column). Statistically significant differences (one-sided Mann-Whitney U-test at 5% significance level) between the DMD and growth cases are highlighted with green shading if the difference is in line with the hypothesized triggers for particle evaporation and with red shading for significant differences towards the opposite direction. For each box, the central mark indicates the median, and the bottom and top edges of the box indicate the 25th and 75th percentiles, respectively. The whiskers extend to the most extreme data points not considered outliers, and the outliers (distance from the top or bottom edge of the box more than 1.5 times the interquartile range) are plotted individually using the 'o' symbols.**







**Figure 5: Comparison of changes in (a, b) temperature, (c, d) wind speed, (e, f) boundary layer height and (g, h) theoretical no-sky radiation between growth hours (blue bars) and the hours when transition into DMD phase occurs (orange bars) separately for summer (months: Apr–Sep, left column) and winter (months: Oct–Mar, right column). For variable X the change ΔX at time t is calculated as the difference between the hourly means at t+1h and t-1h. Values are included in the DMD start category at time t if the DMD start time is found at t±0.5h. Statistically significant differences (one-sided Mann-Whitney U-test at 5% significance level) between the DMD start and growth cases are highlighted with green shading if the difference is in line with the hypothesized triggers for particle evaporation and with red shading for significant differences towards the opposite direction. For each box, the central mark indicates the median, and the bottom and top edges of the box indicate the 25th and 75th percentiles, respectively. The whiskers extend to the most extreme data points not considered outliers, and the outliers (distance from the top or bottom edge of the box more than 1.5 times the interquartile range) are plotted individually using the 'o' symbols.**



## 4.2 NPF footprint areas for growth and DMD phases

In Sect. 3, we speculated that in order to produce apparent shrinkage, the air masses observed during the DMD phase would need be located outside the region of high precursor concentrations at the onset time of NPF. That is, the NPF footprint area (as defined in Sect. 2.3.1) of the particles observed in the DMD phase should lie in the less polluted regions. The fruition of

this condition is studied in Fig. 6.

In Fig. 6a and b, we show the NPF footprint areas for the growth and DMD phases, respectively. In order to illustrate the transition region between these cases more clearly, we show the fraction of the NPF footprint for growth from the total NPF footprint in Fig. 6c. In Fig. 6c, the values are related to the probability of observing growing particles at the measurement

site with respect to the air mass location during NPF start. The NPF footprint area for the growth hours - and the probability of observing an NPF event in the growth phase - show clear resemblance with the areas of high $SO_2$ concentrations, shown in Fig. 6d. By contrast, the DMD footprint - and the probability of observing an NPF event in the DMD phase - are very much focused on the cleaner regions, mainly above the ocean (note the logarithmic axis in Fig. 6a and b). Towards the west, the transition consistently occurs along the coastline where the concentration of $SO_2$ is seen to drop significantly. Towards

north and south, the growth-favoring region extends clearly further consistently with the high concentration region. This implies that the smaller particles observed during the DMD phase are indeed formed in initially less polluted air masses, in line with the first hypothesized condition that would be needed for the apparent shrinkage process (see Sect. 3).

However, in order to produce DMD, lesser growth during transport over the high concentration area would additionally be

required (second hypothesized condition in Sect. 3). The higher wind speeds observed during the DMD phase (Fig. 4c, d) already provide a possible explanation for the lesser growth as they contribute to shorter residence times over the higher concentration area and potentially dilute the concentrations as well. While the behavior of temperature and BLH did not seem to support evaporation (or reduced growth) at the measurement site during the times when DMD was observed to take place, they might still contribute to evaporation or reduced growth at different times during transport. In the following

section, we consider the spatiotemporal variability in the conditions that could modulate aerosol growth during air mass transport and study their effects on the resulting particle size distribution using a Lagrangian single-particle growth model.





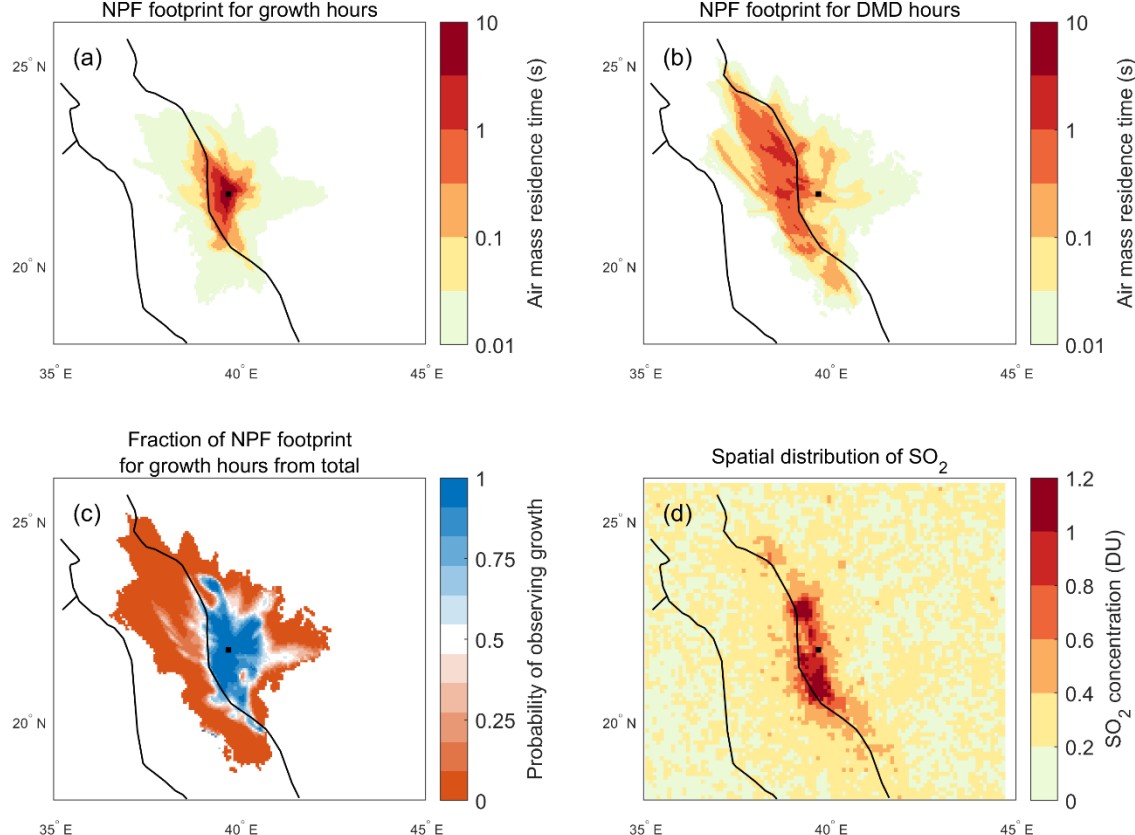

**Figure 6: Average NPF footprint areas (see Sect. 2.3.1) for (a) growth hours and (b) DMD hours. Panel (c) shows the fraction of the (non-averaged) NPF footprint of the growth hours from the total NPF footprint. The result in (c) corresponds to the likelihood of observing growth at the measurement station with respect to the air mass location at the onset time of NPF. A value of 1 means that air masses located in these regions at the onset time of NPF will always result in observed particle growth once transported to the measurement site. Conversely, a value of 0 means that DMD is always observed. In panel (d) the average SO₂ column concentration ((1 DU = 2.69×10^{16} molecules cm^{-2}), obtained from the OMI Level 2 SO₂ Planetary Boundary Layer product (Li et al., 2013), in the surroundings of Hada Al Sham during NPF days is shown.**

## 4.3 Modeling the NPF development with a Lagrangian single-particle model

In order to get a more complete picture of the processes contributing to the DMD events, we develop a Lagrangian single-particle growth model utilizing the available data (see Sect. 2.4) and apply it to model the diameter development of newly formed particles on NPF event days. In short, the model considers the condensation/evaporation of two species, out of which one is completely non-volatile (representing neutralized sulfuric acid) and the other is potentially volatile (representing some low- to semivolatile organic compound of anthropogenic origin). Overall, the model relies on the assumption that the general



spatial distribution of the precursor vapors for both the non-volatile and the volatile species are described by the satellite-retrieved $SO_2$ distribution (shown in Fig. 6d) and that the production of the condensable vapors from the precursors depends linearly on radiation. Additionally, we consider the possible modulating effects of BLH, WS and CS on the vapor concentrations, but both the magnitudes and directions of these effects (i.e. the exponents eBLH, eWS and eCS in Eq. (7))

are treated as unknowns. Due to no measurement data, the true concentrations and properties of the condensing species are also unknown, but the measured particle size distribution constrains the space of possibilities and can be used to infer the more likely effects and conditions. In practise, our approach is to perform a multitude of model runs with varying model configurations in order to find the best-performing description, determined by the model evaluation metrics (described in Sect. 2.4.4). If we then assume this description to be the most likely representation of the real case, we can disentangle the

contributions from evaporation and transport, as well as the factors affecting their occurrence.

### 4.3.1 Finding the best-performing model configuration

The results of our model evaluation are presented in Figs. A3-A6 in the Appendix and in Fig. 7 in the main text. In detail, the model evaluation proceeds by first setting all the exponents eBLH, eWS, eCS in Eq. (7) to zero and the concentration multiplier $f_{\mathrm{vol}}$ and the enthalpy of vaporization $\Delta h_{\mathrm{vol}}$ of the volatile component to 1 and 80 kJ mol[-1], respectively. We then

vary each of the exponents (in the order stated above) and inspect the model performance based on the evaluation metrics over the full possible range of concentrations of the non-volatile component (from 0 to values where the non-volatile condensation alone overestimates the growth; represented with the range of $f_{\mathrm{non}} = 0...1$) and saturation vapor pressures of the volatile component (from fully volatile to non-volatile, with a specific focus on the low- to semivolatile range; represented with the range of $P_{\mathrm{sat,vol}}(T_0) = 0, 1e\text{-}9...1e\text{-}5, \infty$ Pa). From Figs. A3, A4 and 7, that show the performance with varying

eBLH, eWS and eCS, respectively, we can see that any single evaluation metric displays improvement/deterioration over the full range of $f_{\mathrm{non}}$ and $P_{\mathrm{sat,vol}}(T_0)$, and that the responses of the different evaluation metrics correlate with one another with the varying exponents. This makes the selection of the best-performing exponents straightforward, and the obtained values are eBLH = 0.5, eWS = 0 and eCS = -0.5. We will next discuss the implications of the obtained exponents and then inspect the effects of varying $f_{\mathrm{vol}}$ and $\Delta h_{\mathrm{vol}}$, which result in more subtle changes in the model performance (Figs. A4 and A5).





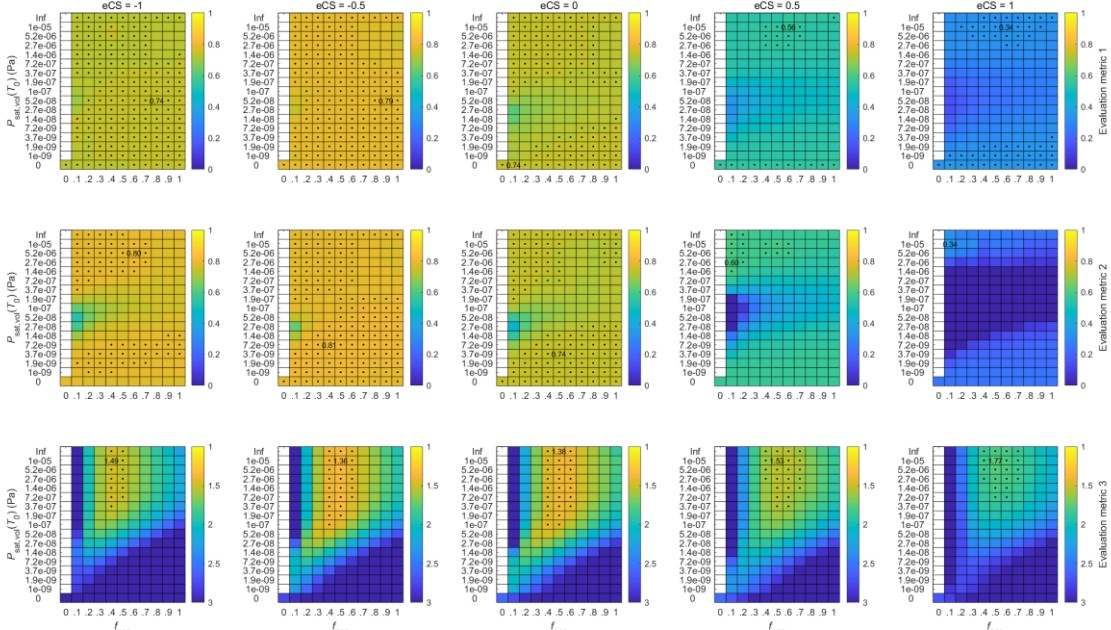

**Figure 7: Evaluation of the model performance with different exponents for the condensation sink term (eCS) when eBLH = 0.5, eWS = 0, $f_{vol}$ = 1 and $\Delta h_{vol}$ = 80 kJ mol$^{-1}$. Each column of panels contains the performance matrices of the three evaluation metrics (see Sect. 2.4.4 for explanations of the metrics) as a function of the concentration multiplier for the non-volatile compound ($f_{non}$) and the saturation vapor pressure of the volatile compound ($P_{sat,vol}(T_0)$) with a specific eCS value. On the first two rows, higher correlation values of the evaluation metrics indicate better model performance, while on the last row, lower deviation correspond to better results (better results are always towards the yellow colors). In each panel, the value of the evaluation metric for the best model performance is shown in numbers. Values close to the best (difference less than 0.015 in the correlation values and less than 5% in the deviation value) are highlighted with black dots. Each model run, resulting in a single data point for each of the evaluation metrics, comprises 138 NPF event days.**

The positive exponent for the BLH term produces higher vapor concentrations with increasing BLH. This is contrary to the expected effect of increasing dilution by increasing boundary layer height, but matches the observations of generally lower BLH during DMD events, found in Sect. 4.1. Positive exponents for the BLH term from could result from the residual layer containing similar or higher amounts of precursor vapors than the mixing layer or by the CS being diluted relatively more with increased vertical mixing than the precursor vapors. Such situation could potentially arise from the consistent sea breeze circulation at the study region (Parajuli et al., 2020) and elevated stack emissions over the shallow nocturnal boundary layer (Ukhov et al., 2020b) with additional contributions from the higher dry deposition velocities of gases compared to particles in the ~100 nm range that control the CS (Seinfeld and Pandis, 2016). In the United Arab Emirates, Kesti et al. (2022) report indications of elevated layers with high SO$_2$ concentrations as the surface concentrations often increase with entrainment from the residual layer. Recent modeling studies over China have also suggested that more favorable conditions for new particle formation and growth, including higher concentrations of precursor vapors and oxidants as well as a lower sink and



temperature, might exist higher up in the atmosphere, causing increased vertical mixing to result in enhanced particle formation at the surface (Lai et al., 2022a;Lai et al., 2022b). However, in our case the positive eBLH could also be related to the diurnal variation in emissions, which are not considered in the model, through the correlation of BLH and temperature, and temperature and emissions in a region where energy consumption responds to the need for air conditioning (Ukhov et al., 2020b). An increase in the BLH value during transport may also signal the arrival of an air mass, initially located over sea, to land areas, where the magnitude in the diurnal variation of BLH is much stronger. In such case, the positive eBLH could stem from the concentration gradient between land and sea being stronger in reality than described by the satellite-retrieved distribution used here. However, we also briefly tested higher exponents for the [SO$_2$] term, which result in an increased contrast between the high and low concentration areas, but this did not yield improved results. While several possible explanations for the positive eBLH exist, we are unable to pinpoint the underlying effects.

Higher wind speed was found to favor the occurrence of DMD events in Sect. 4.1. The obtained zero exponent for the WS term here suggests that this connection is related to transport effects rather than dilution effects. This is because the transport related effects of wind speed are inherently included in the Lagrangian framework, via the use of air mass trajectories, and thus their contribution is not expected to show up in the exponent here (whereas the dilution effects would be). While the wind conditions will certainly modulate the precursor concentrations to some degree, it seems that the major influence of wind speed comes from its effect to the travel time over the high concentration area.

The negative exponent for the CS term is contrary to the expectation of decreasing concentrations of condensing vapors with increasing sink. However, it is not completely unexpected at this particular site, as we previously found particle formation and growth rates to be positively correlated with the CS (Hakala et al., 2019). We believe this to reflect the common anthropogenic sources of NPF precursor vapors and large (possibly primary) particles that mainly control the CS. That being said, faster particle growth over the anthropogenically active areas should ideally be already accounted for by the [SO$_2$] field without the need of additional input from the observed CS. As such, the improved model performance when using eCS = -0.5 might act to correct some of the errors that undoubtedly arise from the highly simplistic description of the precursor vapor distribution, as well as the representation of the air mass movements by the trajectories. Since the measured CS is expected to increase as a result of particle growth, it might provide direct input on the intensity of condensation that is not otherwise captured by the model.

In terms of changing the concentration multiplier ($f_{vol}$) and the enthalpy of vaporization ($\Delta h_{vol}$) for the volatile component, the responses in the overall model performance are much less pronounced (Figs. A5 and A6). Varying either of these does not significantly affect the best model performance within any metric, but mostly shifts the regions in which similar model performance is found. The shifting of the regions is expected, as largely analogous conditions are obtained by combining higher concentrations of the volatile component with higher saturation vapor pressures, and higher enthalpies of vaporization



with lower saturation vapor pressures (the slope of the saturation vapor pressure curve with respect to temperature increases with increasing $\Delta h_{vol}$, and thus in order to reach similar $P_{sat,vol}(T)$ values at a specific $T > T_0$, a lower initial $P_{sat,vol}(T_0)$ is needed). The condition are, however, not exactly the same as both the $f_{vol}$ and $\Delta h_{vol}$ can affect the timing and intensity of the condensation-evaporation dynamics over the diurnal cycle, with higher $f_{vol}$ essentially promoting more intense condensation

and evaporation, and higher $\Delta h_{vol}$ making the saturation vapor pressure more sensitive to the diurnal variation in temperature. From Fig. A5 we can see that while the best model performance is not affected by the choice of $f_{vol}$, the worse-performing regions deteriorate with increasing $f_{vol}$. This suggests that strong contributions from the volatile component are not favorable for the model performance. The very low overall model sensitivity to variations in $\Delta h_{vol}$ (Fig. A6) in turn signals that the changes in the concentrations of the volatile component dominate over the temperature-dependent changes in the saturation

vapor pressure over the diurnal cycle. As such, the choice of the value of $\Delta h_{vol}$ seems largely inconsequential in terms of further analysis while choosing the value of $f_{vol}$ still requires further considerations.

In the next section, we will focus on the model performance between individual model runs with specific concentration multipliers of the non-volatile component (determined by $f_{non}$) and saturation vapor pressures of the volatile component

$P_{sat,vol}(T_0)$. The discussion is based on the results shown in the second column of Fig. 7, which are considered to represent the best performing model configuration: eBLH = 0.5, eWS = 0, eCS = -0.5, $f_{vol}$ = 1 and $\Delta h_{vol}$ = 80 kJ mol$^{-1}$. As mentioned, the value of $f_{vol}$ is still discussed further.

### 4.3.2 Contributions from the volatile and non-volatile components

In the previous section, we found the best overall model performance (with any $f_{non}$ and $P_{sat,vol}(T_0)$) when eBLH = 0.5, eWS =

0, eCS = -0.5, $\Delta h_{vol}$ = 80 kJ mol$^{-1}$ (and $f_{vol}$ = 1) (Fig. 7). Here we will discuss the results in terms of varying contributions from the volatile and non-volatile components, determined by $f_{non}$ and $P_{sat,vol}(T_0)$ respectively. In general, we find quite low sensitivity to changes in $f_{non}$ and $P_{sat,vol}(T_0)$ in the first two model evaluation metrics (Fig. 7, first two rows), which depend only on the shape (and not the absolute values) of the modeled diameter development. However, a band of poorer performance that covers a specific $P_{sat,vol}(T_0)$ range and extends towards higher volatilities with increasing $f_{non}$, can be seen

especially in the second evaluation metric. The same region was found to be more pronounced with higher $f_{vol}$ (Fig. A5). In this region, the condensation-evaporation dynamics of the volatile component are particularly sensitive to the saturation vapor pressure. This can be inferred from the fact that above and below this region the behavior resembles that of a completely volatile and non-volatile vapor, respectively, as seen from the similar values to the cases of $P_{sat,vol}(T_0) = \infty$ and 0. This shows that some combinations of $f_{non}$ and $P_{sat,vol}(T_0)$ lead to semivolatile dynamics that do not match the observed shape

of the particle diameter development, while a wide range of conditions produce results close to the best obtained values.

The third evaluation metric (Fig. 7, third row), which reflects the numerical agreement between the observed and modeled diameter values, naturally places much more strict limits to the choice of $f_{non}$ and $P_{sat,vol}(T_0)$. With no contribution from the



volatile component ($P_{sat,vol}(T_0) = \infty$), we find a local maximum in the performance with $f_{non} = 0.5$. Note that this local maximum is very close to the global maximum found at $f_{non} = 0.5$ and $P_{sat,vol}(T_0) = $ 1e-5 Pa, which we will later show to represent practically non-volatile condensation with negligible contribution from the volatile component. With $f_{non} > 0.5$, the contribution from the non-volatile component already exceeds the typically observed particle diameters and thus the model performance decreases monotonically with increasing contribution from either the non-volatile or the volatile component. Therefore, the region of interest lies in the values of $f_{non} \leq 0.5$, where decreasing contribution from the non-volatile compound needs to be compensated with an increasing contribution from the volatile compound. However, when looking at performance of the third evaluation metric in Fig. 7, we quickly come to realize that increasing need for the volatile compound (decreasing $f_{non}$) simultaneously results in worse model performance, regardless of the chosen $P_{sat,vol}(T_0)$. This is related to the fact that in order to get a quantitatively suitable contribution from the volatile compound when $f_{vol} = 1$, its $P_{sat,vol}(T_0)$ needs to lie in the range where the condensation is clearly modulated by the saturation vapor pressure; with very high $P_{sat,vol}(T_0)$, no condensation will occur and the resulting particles will be too small (especially when $f_{non} \leq 0.3$), whereas with very low $P_{sat,vol}(T_0)$, the condensation will approach that of a non-volatile compound, which will produce too large particles if $f_{vol} = 1$. This is because a non-volatile compound alone already produces too large particles with $f_{non/vol} > 0.5$, as discussed above. However, this same semivolatile regime, where the volatile component needs to lie, was found to worsen the model performance especially in terms of the second evaluation metric, indicating a poorer match in the shape of the modeled particle diameter development with the observations.

The "problem" of having to place the volatile component in the semivolatile regime can, of course, be circumvented by using an $f_{vol}$ value lower than 0.5. However, in this case the best model performance is found when the volatile component simply behaves as another non-volatile one, as can be seen from the case of $f_{vol} = 0.2$ in Fig. A5. In this case, no evaporation is possible. The $f_{vol}$ range between 0.2 and 0.5 was further studied with small increments of 0.05 to confirm that the best performance is never found in the semivolatile regime (Fig. A7). Thereby, any significant contribution from a truly semivolatile component does not aid in reproducing the observed diameter development with our model. This suggests that the evaporation of a volatile compound is unlikely to be an important contributor to the DMD events.

Based on the results and discussion above, we deem the case of $f_{non} = 0.5$ and $P_{sat,vol}(T_0) = $ 1e-5 Pa (with eBLH = 0.5, eWS = 0, $\Delta h_{vol} = 80$ kJ mol$^{-1}$, $f_{vol} = 1$; Fig. 7) representative of the best-obtainable results by our model. The strong correlation between the modeled and observed diameters and their changes ($r = 0.79$ and $r = 0.80$, evaluation metrics 1 and 2, respectively) and the modeled particle diameters being on average within a factor of 1.36 from the observed ones (evaluation metric 3) indicate good performance of the model. In the next section, we will illustrate the model-produced particle diameter development and address the causes of the DMD events more specifically.





### 4.3.3 Contributions from evaporation and transport

In Figs. 8 and 9, we illustrate the development of the observed and modeled mean particle diameters together with the modeled contributions from true diameter changes and transport effects (see Sect. 2.4.3) for two cases chosen based on Fig. 7. In Fig. 8, we show the case of the best model performance ($f_{non}$ = 0.5 and $P_{sat,vol}(T_0)$ = 1e-5 Pa), while in Fig. 9 we illustrate an example of a case where the non-volatile component alone is clearly insufficient in explaining the observed particle diameters (here $f_{non}$ = 0.3, $P_{sat,vol}(T_0)$ = 5.2e-8 Pa). In the latter case, a clear contribution from the volatile component is needed, which results in a poorer model performance.

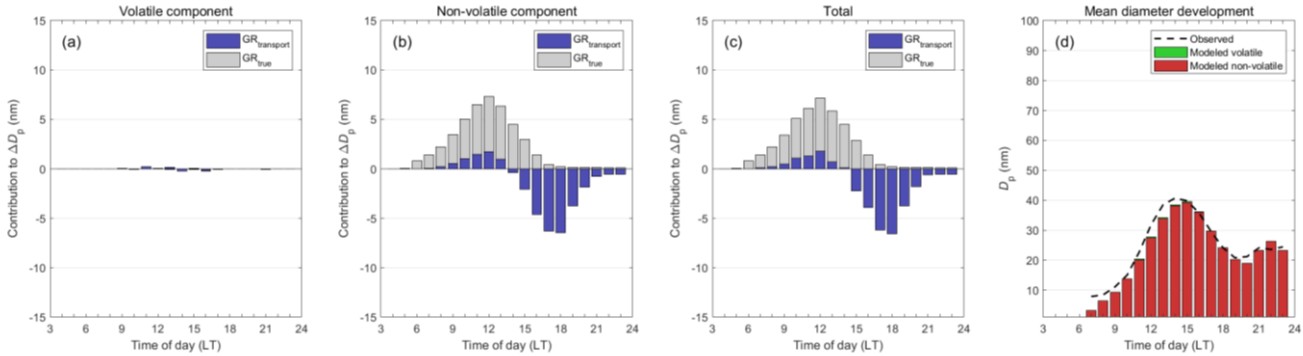

**Figure 8: Mean contributions to modeled diameter changes by (a) the volatile component, (b) the non-volatile component and (c) both the volatile and non-volatile components when using the best performing model configuration (eBLH = 0.5, eWS = 0, eCS = -0.5, $\Delta h_{vol}$ = 80 kJ mol$^{-1}$, $f_{vol}$ = 1, $P_{sat,vol}(T_0)$ = 1e-5 Pa and $f_{non}$ = 0.5). The diameter contributions are separated into contributions from true diameter changes (GR$_{true}$; condensation, evaporation) and transport (GR$_{transport}$), described in Sect. 2.4.3. (d) Observed and modeled mean particle diameter. All panels contain data from 138 NPF event days. In panel (d), only hours when the observed mode diameter is defined are included.**

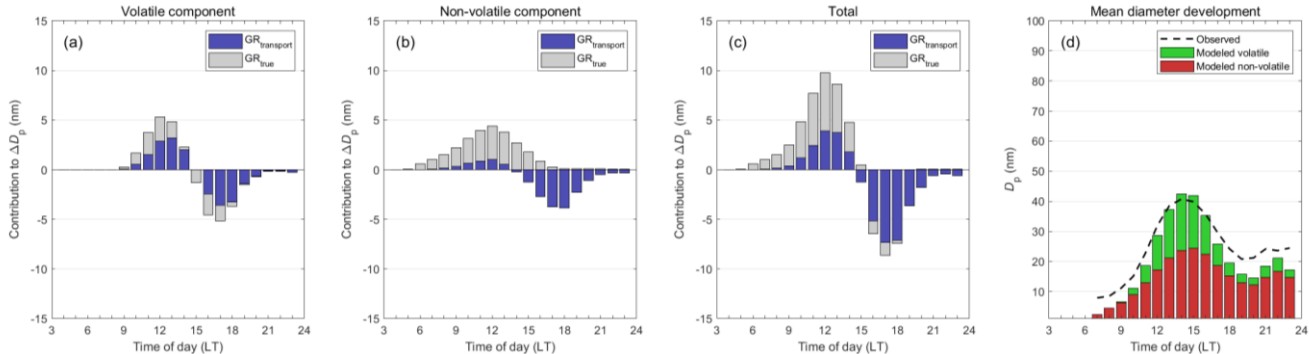

**Figure 9: Same as Fig. 8 but with $f_{non}$ = 0.3 and $P_{sat,vol}(T_0)$ = 5.2e-8 Pa.**





From Fig. 8 we can see that in the case of the best model performance, the contribution from the volatile component is on average negligible. Despite this, the mean diameter development - including the DMD behavior - is reproduced remarkably well. With the non-volatile component, no evaporation is possible and thus the DMD events are caused by the apparent shrinkage process. In this process, the decreasing size of the observed particles results from differing conditions during transport, as directly shown by the strongly negative contributions from $GR_{transport}$ to the diameter changes especially in the afternoon (Fig. 8b or c). In addition to the negative contribution from transport, we can also see a smaller positive contribution during the earlier growth hours. This is caused by the generally increasing concentrations of the precursors from the immediate vicinity of the measurement site towards the high emission areas along the coast (Fig. 6d). Therefore, according to the model results, calculating particle growth rates directly from the observed size distribution would typically give an overestimate of the real condensational growth at the measurement site during the early hours. However, a much more severe misjudgement would be made if the DMD events were similarly interpreted to represent net evaporation of particles, since the true diameter changes of individual particles are continuously positive.

If we choose a worse-performing model configuration where significant contribution from the volatile component is needed, net evaporation of the volatile component is also found to contribute to the diameter changes (Fig. 9a). The evaporation peaks around 16 LT, when the photochemical production rate has significantly dropped from its midday maximum while the temperature is still around its highest values (see Fig. 4). However, while evaporation can be significant for the volatile component, in the total contributions some of the evaporation is compensated by simultaneous condensation of the non-volatile compound, and overall the DMD is still predominantly caused by transport in the displayed case. The relative contributions of the different terms are obviously dependent on the chosen value of $f_{non}$, but we remind that the overall agreement between the observations and the model was found to weaken with decreasing $f_{non}$. With lower $f_{non}$ and higher contribution from a semivolatile compound, both the early morning and late evening diameters become progressively more underestimated while the daily maximum diameters are overestimated due to the strong diurnal variation in the condensation-evaporation dynamics of the semivolatile component. Such variation is not in line with the observed diameter development, making significant contributions from volatile compounds and evaporation seem unlikely.

### 4.3.4 Hypotheses vs model results

Lastly, we comment on the hypothesized causes of the DMD events presented in Sect. 3 based on the obtained model results. We found practically non-volatile condensation to best explain the observations. This suggests that the DMD events are caused by the apparent shrinkage process, enabled by differing conditions during transport. In Fig. 10, we illustrate these differing conditions, in terms of the concentration of the non-volatile component, and the contributing factors for the daily largest and smallest modeled particles using the best-performing model configuration. In agreement with the first hypothesized condition for the apparent shrinkage (Sect. 3), we found that the particles observed during the DMD phase are located outside the high concentration region around the onset of NPF (Fig. 6). This can also be seen from Fig. 10b and



identified as a major contributor to the DMD events, since most of the diameter difference (reflected by the integral of the concentration difference in Fig. 10c) between the daily largest and smallest particles results from the DMD air mass being located in the lower concentration area until 14 LT. In addition to the lower [SO$_2$], the lower BLH in the DMD air mass also contributes to the slower initial growth in our model.

We are not able to comment whether in reality the later NPF start or the slower growth in the initially cleaner air masses (subpoints of the first hypothesized condition in Sect. 3) is the main cause of the initial setback in the diameter development. This is because in our model any non-zero value of radiation will simultaneously initiate particle growth everywhere, regardless of the SO$_2$ concentration. Thus our model only considers the first option, while in reality, slow-growing particles

formed far away from the measurement site could be lost to coagulation before they reach our observations. Therefore, particles formed later in the day closer to the measurement site might also be relevant.

In Sect. 3, we stated that after the initial diameter setback, the particles in the DMD air mass need to grow less during the transport over the high concentration area, in order to not catch up with the earlier-observed particles despite the extended

growth time. We presented several factors that could potentially lead to such conditions. Since the practically non-volatile case was found to perform best, the particle growth is not hindered by the effects of temperature on volatility in our model. Further, we found a zero exponent for the WS term and a positive exponent for the BLH term, meaning that diluted precursor concentrations due to increasing wind or boundary layer height are also not causing the reduced growth. In fact, the DMD air masses seemed to be related to lower BLH and thus letting the lower BLH contribute towards lower vapor

concentrations was found to benefit our model. The lower BLH, which was already found to contribute to the initial diameter setback, is also partly responsible for the reduced growth over the high concentration area (Fig. 10b, c). While a minor contribution is also found from the CS term, the main reason for the reduced growth over the high concentration area in the DMD air mass is the significantly reduced photochemical production of the non-volatile vapor in the afternoon due to the reduced radiation. This becomes more evident if Fig. 10 is plotted with the horizontal axis in hours relative to the

observation time of the arriving particles, instead of the time of day (shown in Fig. A8).

We also find that the DMD air masses, in which the smallest particles in the afternoon are observed, travel over the high concentration area slightly faster, with the mean residence times being 8 and 7 hours for the air masses containing the daily largest and smallest particles, respectively. However, such a small difference should not be a significant contributor to the

overall diameter difference, especially when considering that the concentration gradient is in reality less pronounced than illustrated in the exaggerating schematic (Fig. 10b vs Fig. 2). This also suggests that the clear difference found in the wind speeds between the DMD and growth phases (Fig. 4c, d) could simply be related to an earlier onset of DMD on the overall windier days, as the air masses that resided in the low concentration regions around NPF start would be transported to the measurement site sooner. Overall higher wind speeds will also result in more pronounced transport effects. This could



possibly skew our identification of the DMD cases towards stronger winds, as clear signals of DMD were required in the classification.

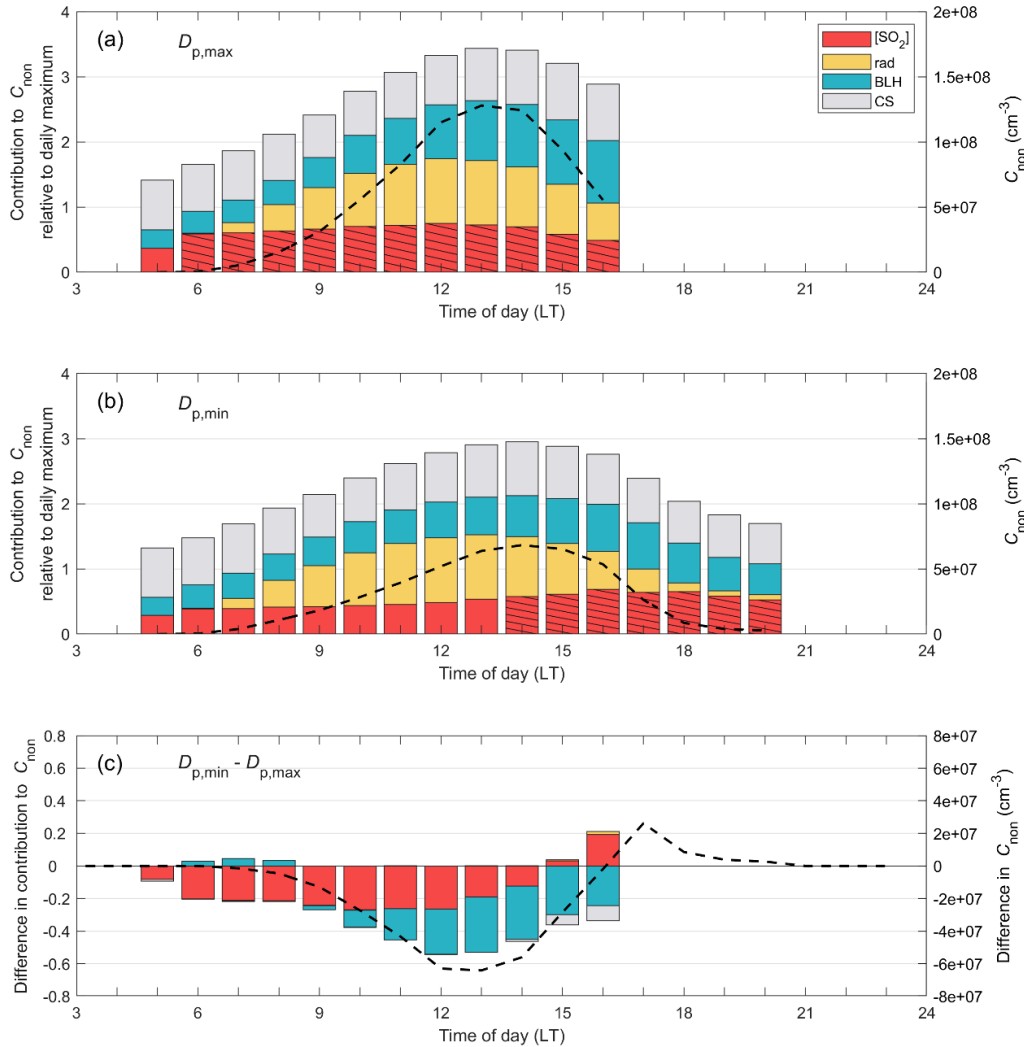

**Figure 10: (a, b) Left axis, bars: Mean contributions of the different factors (relative to daily maximum) affecting the calculated concentration of the non-volatile component ($C_{non}$) during air mass transport to the measurement site. Right axis, dashed black line: The concentration of the non-volatile component (note that the concentration responds to the product of the different factors). In panel (a), we show the mean conditions resulting in the daily largest modeled particle ($D_{p,max}$) and in panel (b) the mean conditions resulting in the smallest modeled particle in the evening ($D_{p,min}$) during the time when the NPF related mode is still observed in the measurements. Only the most typical arrival times for the largest and smallest particles are considered (n > 10; hours 13-16 for the largest and 17-20 for the smallest particles). The dashed bars mark the times when the air masses are typically located over the high concentrations area (here defined as [SO₂] > 0.55 DU). (c) The difference between the conditions resulting in the smallest and largest particles ((b)-(c)). The concentration line in (c) is shown also after 16 LT to display the small positive contribution from the additional growth time compared to the negative contribution resulting from less favorable conditions for growth during transport. The difference in the sizes of the smallest and largest particles at a given time is reflected by the integral of the concentration difference in (c) up to that time.**



## 5. Discussion

Both observational and modeling studies suggest that a very high fraction, up to 80%, of the non-dust $PM_{2.5}$ in the Arabian Peninsula region consist of sulfate (Kesti et al., 2022;Ukhov et al., 2020a;Randles et al., 2017). This results from the high emissions of $SO_2$ (Liu et al., 2018) and low emissions of VOCs (Henze et al., 2008;Alghamdi et al., 2014;Sindelarova et al.,

2014). Therefore, it can be expected that a significant fraction of the ultrafine particles formed in NPF events in this region would also consist of sulfate. Sulfate aerosol does not show significant evaporation at ambient temperatures (Huffman et al., 2009;Xu et al., 2019). In our observations, we often find drastic decreases in particle diameters, e.g., from 80 nm to 20 nm. If evaporation alone was responsible for such diameter changes, 98 % of the particle mass would have to evaporate. This is in clear contradiction with the expected role of sulfate in the region and thus supports our results of apparent shrinkage being

the main cause of the DMD events.

Hada Al Sham is surrounded by the Sahara Desert and the Red Sea in the west and the Arabian Desert in the east, with populated areas concentrated on a narrow band along the coast of the Red Sea. With the mostly barren surroundings, the sources of NPF precursor vapors are highly localized and practically limited to anthropogenic sources (assuming that marine

emissions of e.g. iodine and dimethyl sulfate are of secondary importance compared to the strong anthropogenic emissions). This creates a distinct contrast in the NPF characteristics between the nearby and further-away areas, and the presence of this contrast can be regularly experienced in Hada Al Sham due to the consistently effective transport conditions created by the sea breeze. Since the apparent shrinkage process depends on the presence of such conditions, this could explain why the DMD events are so common in Hada Al Sham.

Many of the other DMD events reported in the literature are also observed in the vicinity of urban or coastal areas (Alonso-Blanco et al., 2017), where clear emission and concentration gradients are expected. However, most of the measurement locations could potentially be more homogeneous in terms of NPF precursor vapors due to the widespread urbanized areas in Europe and East Asia and the non-urban areas often covered by some form of vegetation. Biogenic VOCs can be a

significant contributor to particle growth even in urban areas (Guo et al., 2012;Huang et al., 2016). Thus, on urban sites surrounded by vegetation, the transition from local anthropogenic NPF to more regional NPF, controlled mostly by biogenic emissions, might occur without any striking changes in the PNSD development. For example, Huang et al. (2016) compares two NPF events in Nanjing, one influenced mainly by anthropogenic and the other by biogenic emissions, and reports similar growth rates in both cases despite the differing source regions. Such effects could contribute to the lower frequency of DMD

events found on other sites. In addition, the overall frequency of NPF is typically clearly lower in most sites around the globe compared to Hada al Sham (Nieminen et al., 2018;Hakala et al., 2019), making the exotic exceptions that much more unlikely to come by.

Regarding the interpretation of our modeling results, we wish to point out that while the best model performance was found with practically only the non-volatile component, which was described to represent neutralized sulfuric acid, we do not mean to claim that only neutralized sulfuric acid would be responsible for the growth of the particles. As seen in Fig. A5, very similar results can be obtained with lower contribution from the non-volatile component with a suitable amount of other low-

volatility (organic) vapors, e.g. $f_{non} = 0.3$ and $f_{vol} = 0.2$. Similar result could also be obtained with higher volatility vapors if their particle face activities were high or if e.g. oligomerization suppresses their evaporation.

We acknowledge that our modeling results, and thus also the conclusions drawn from them, are based on a large number of simplistic assumptions, which are made due to either lack of observational data or more sophisticated approaches being

outside the scope of this manuscript. Below we discuss some of the aspects that could potentially affect the conclusions drawn from our modeling results.

As the explanations of the best performing exponents for the CS and BLH terms (eCS = -0.5 and eBLH = 0.5) are somewhat ambiguous, and since the CS provides direct input from the observed PNSD, we comment on the model-observation

agreement with varying $f_{non}$ and $P_{sat,vol}(T_0)$ when eCS = 0 and eBLH = 0 (the results can be seen in Fig. A3, middle column). With this configuration, the model performance decreases less with decreasing $f_{non}$ and increasing contribution from the volatile component but the best results, in terms of metric 3, are obtained with only the non-volatile compound included. While the model performance is naturally worse than in the best case, the mean diameter development is still reproduced relatively well. The cases where significant contributions from the volatile component are needed (e.g. $f_{non} = 0.3$, $P_{sat,vol}(T_0) =$

1.4e-8 Pa) also remain qualitatively similar, with transport still being the main cause of DMD. Therefore, our conclusion about the unlikely role of semivolatile evaporation as the cause of the DMD events is not dependent on the chosen exponents for the CS and BLH terms.

In order to further test the robustness of our results and to disentangle the reasons for the overall poorer performance with the

volatile component, we performed test runs where the factors affecting the evaporation rate of the volatile component in our model were turned on or off one at a time. These are the effects of temperature on the saturation vapor pressure, particle size (Kelvin effect) and the molar fraction of the volatile component (Raoult's law). We found that including the temperature-dependence of the evaporation rate and the Raoult's law improves the agreement between the model and observations, while the Kelvin effect weakens it. However, even when neglecting the Kelvin effect, we were unable to improve (or match) the

results of purely non-volatile condensation (when $f_{non} = 0.5$) by applying $f_{non} < 0.5$. Therefore, our results are not dependent on some of the main assumptions regarding the evaporation rate of the volatile component.



While the precursors of the organic aerosol, represented by the volatile compound in our model, are likely to be of anthropogenic origin, their spatial distribution might differ from that of the $SO_2$ depending on the relative emission intensities of VOCs and $SO_2$ from different sectors (e.g. energy, industry, residential, transport) and the geographical distribution of these sectors. Although not explicitly included for either of the components, the diurnal variation in the emissions of the precursors might also be different. Concerning the production of the condensable vapors, chlorine might play a meaningful role in the VOC oxidation in this region in addition to OH (Bourtsoukidis et al., 2019), which is primarily accounted for in the model. We did not attempt to account for such factors.

We did, however, consider the unlikely scenario where the implemented Lagrangian framework and the assumed spatial distribution of the precursors would be sending completely false signals to the aerosol growth modeling, by performing similar model evaluation as in Sect. 4.3.1 but with using stationary air masses. In this case, no transport effects are possible and only evaporation can produce the DMD events. Quite expectedly, this setup resulted in considerably weakened model performance, highlighting the important role of the transport-effects and the usefulness of the simplistic approach in describing the spatial distribution of the precursors.

## 6. Summary and conclusions

In this study, we focused on investigating the cause of the frequently observed DMD events in Hada al Sham. We considered two fundamentally differing processes as the possible explanations, namely, particle evaporation and apparent shrinkage. In the latter process, no actual reduction in the sizes of individual particles would occur, as the DMD events would be caused by consecutive transport of less-grown particles to the observation site after the more-grown ones.

We first compared the meteorological conditions between the DMD and non-DMD days as well as those between the DMD and growth phases. While several other studies suggest that DMD events could be caused by changes in local meteorological conditions that initiate particle evaporation, we did not find consistent evidence supporting such conclusions.

Estimating the source areas of the particles related to the NPF events indicated that the air masses, in which the DMD events are observed, were consistently located outside the region of strong anthropogenic influence around the start time of NPF. This finding supports the apparent shrinkage process being the cause of the DMD events, since the initial residence in a lower precursor environment could cause a needed lag in the particle diameter development compared to the particles in a higher precursor environment.

The dynamics of the DMD events were studied further using a Lagrangian single-particle growth model. The Lagrangian framework is essential for studying the apparent shrinkage process, as the process relies on particles observed at different

times having undergone different conditions during their transport to the measurement site. In addition, the model also allowed us to address the possible role of evaporation more thoroughly. In our model, we consider the condensation/evaporation of two species, out of which one is completely non-volatile while the other is potentially volatile and thus able to condense or evaporate depending on the prevailing conditions. Our main assumptions concerning the concentrations of these species were that both of them are predominantly produced photochemically from precursor vapors whose spatial distribution is described by the satellite-retrieved $SO_2$ concentration in the study region.

Despite the additional degrees of freedom provided by the potentially volatile component, we found the best agreement between the modeled and observed particle diameters when practically only the non-volatile component was responsible for the diameter changes. Our results clearly demonstrate that the observed DMD events can be produced by the apparent shrinkage process with physically sensible assumptions, without the need for particle evaporation. Further, the deteriorating model-observation agreement with increasing contributions from a semivolatile compound suggest that evaporation is, in fact, unlikely to play a significant role in our observations.

If the DMD events in Hada al Sham are indeed caused by the apparent shrinkage process, the relevance of the NPF events towards CCN production is greatly enhanced in this region compared to evaporation being the cause. This is simply because in the apparent shrinkage process, none of the particles that reach large-enough sizes to become climatically active are inactivated in the future, barring losses to coagulation or deposition. In addition, even the smaller, slow-growing, particles observed during the DMD events can continue their growth towards larger sizes.

Similar transport-related effects to those found in this study might also contribute or cause the DMD events found in other environments. Apart from the study by Kivekäs et al. (2016), we are not aware of studies explicitly including spatially varying growth rates in the analysis of DMD events. In a broader context, transport-related effects are often either assumed minor or neglected altogether in the analysis of aerosol growth rates. While our observations are likely to represent a rather extreme case, they highlight the importance of considering the effects arising from horizontal heterogeneities and transport on the interpretation of fixed-point ambient observations, especially in relation to NPF events.

**Data availability**

Data used in this study are available from the corresponding author upon request (simo.hakala@helsinki.fi).



## Author contribution

Conceptualization: SH, PP. Formal Analysis: SH. Funding acquisition: SH, MK, TP, TH, MIK, MAA, PP. Investigation: HL, APH, KN, MIK. Methodology: SH. Project administration: HL, APH, TP, TH. Software: VV. Supervision: PP, MK, VMK. Visualization: SH. Writing – original draft: SH. Writing – review & editing: SH, VV, HL, APH, KN, JK, VMK, MK, TP, TH, MIK, MAA, PP.

## Competing interests

The authors declare that they have no conflict of interest.

## Acknowledgements

The authors wish to acknowledge CSC – IT Center for Science, Finland, for generous computational resources and Anu-Maija Sundström for their contributions towards utilizing the OMI satellite data. TH acknowledges The Eastern Mediterranean and Middle East Climate and Atmosphere Research (EMME-CARE) project, which received funding from the European Union's Horizon 2020 Research and Innovation Programme (Grant Agreement Number 856612) and the Government of Cyprus. The sole responsibility of this publication lies with the authors.

## Financial support

This study was funded by the Deanship of Scientific Research (DSR, grant no I-122-430) at King Abdulaziz University (KAU), the Academy of Finland (ACCC Flagship, project 337549; Centre of Excellence program, projects 272041 and 307331; Profi 3 program, project 311932; Academy professorship, project 302958; and projects 325656 and 325647), the European Research Council (ERC) under the European Union's Horizon 2020 research and innovation program (ATM-GTP, grant agreement 742206; FORCeS, grant agreement 821205), the European Commission Horizon Europe project FOCI, "Non-CO2 Forcers and Their Climate, Weather, Air Quality and Health Impacts (project 101056783) and the Doctoral Programme in Atmospheric Sciences at the University of Helsinki (ATM-DP).

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

## Appendix A

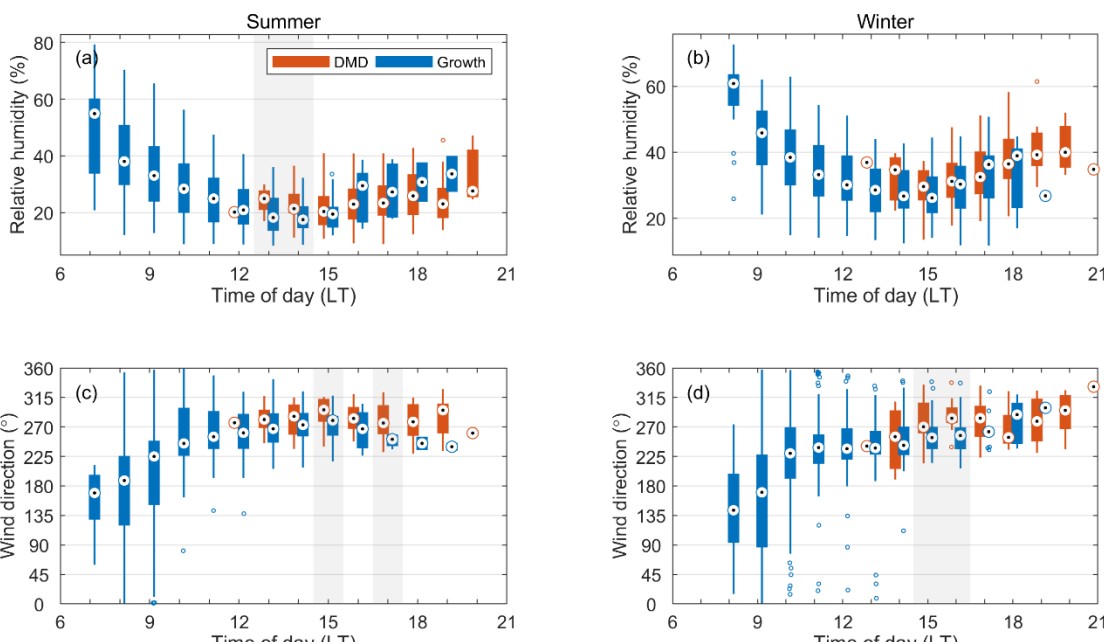

Figure A1: Comparison of hourly mean (a, b) relative humidity and (c, d) wind direction between NPF events in growth phase (blue bars) and DMD-phase (orange bars) separately for summer (months: Apr–Sep, left column) and winter (months: Oct–Mar, right column). Statistically significant differences (two-sided Mann-Whitney U-test at 5% significance level) between the DMD and growth cases are highlighted with grey shading. For each box, the central mark indicates the median, and the bottom and top edges of the box indicate the 25th and 75th percentiles, respectively. The whiskers extend to the most extreme data points not considered outliers, and the outliers (distance from the top or bottom edge of the box more than 1.5 times the interquartile range) are plotted individually using the 'o' symbols.



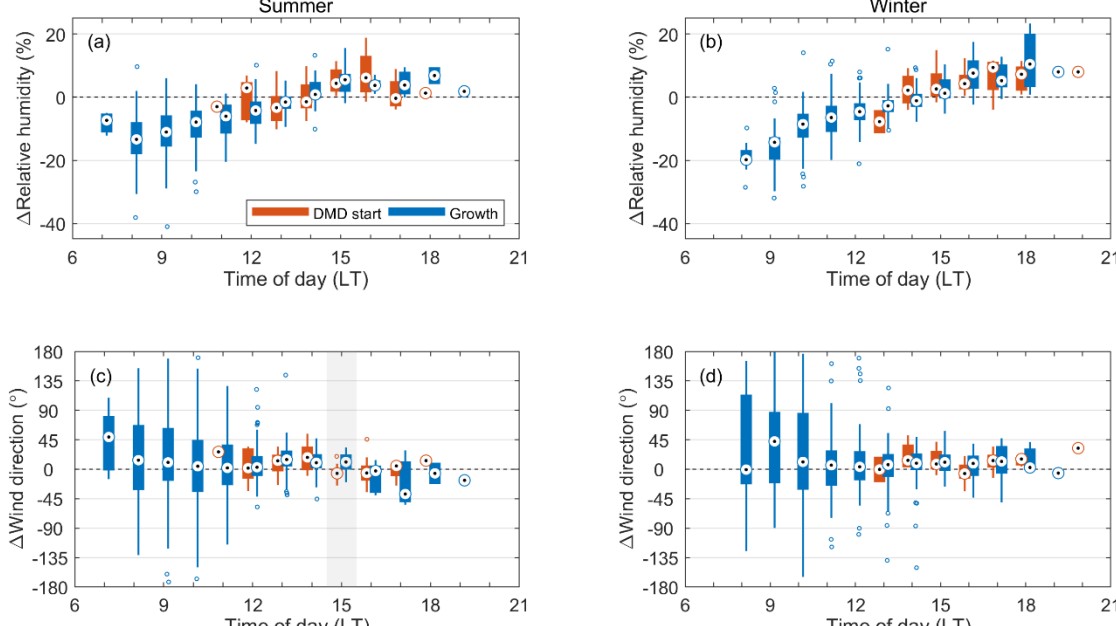

**Figure A2: Comparison of changes in (a, b) relative humidity and (c, d) wind direction between growth hours (blue bars) and the transition into DMD-phase (orange bars) separately for summer (months: Apr–Sep, left column) and winter (months: Oct–Mar, right column). For variable X the change ΔX at time t is calculated as the difference between the hourly means at t+1h and t-1h.**
5  **Values are included in the DMD start category at time t if the DMD start time is found at t±0.5h. Statistically significant differences (two-sided Mann-Whitney U-test at 5% significance level) between the DMD start and growth cases are highlighted with grey shading. For each box, the central mark indicates the median, and the bottom and top edges of the box indicate the 25th and 75th percentiles, respectively. The whiskers extend to the most extreme data points not considered outliers, and the outliers (distance from the top or bottom edge of the box more than 1.5 times the interquartile range) are plotted individually using the 'o'**
10 **symbols.**





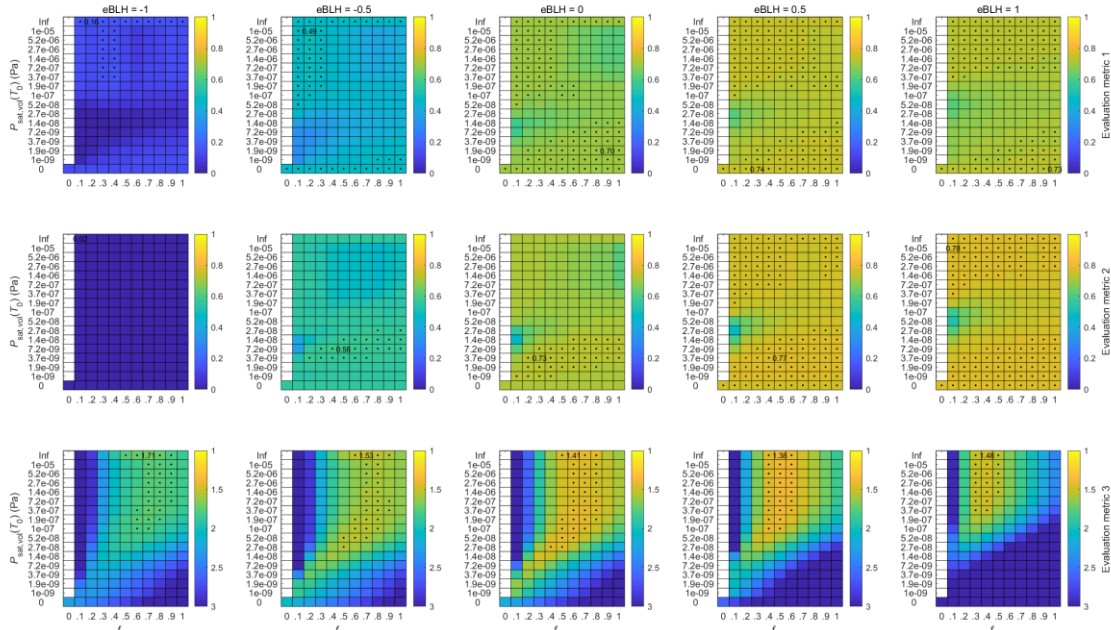

**Figure A3. Evaluation of the model performance with different exponents for the boundary layer height term (eBLH) when eWS = 0, eCS = 0, $f_{vol}$ = 1 and $\Delta h_{vol}$ = 80 kJ mol$^{-1}$. Each column of panels contains the performance matrices of the three different evaluation metrics as a function of the concentration multiplier for the non-volatile compound ($f_{non}$) and the saturation vapor pressure of the volatile compound ($P_{sat,vol}(T_0)$) with a specific eBLH value. On the first two rows, higher correlation values of the evaluation metrics indicate better model performance, while on the last row, lower deviation correspond to better results. In each panel, the value of the best model performance is shown in numbers. Values close to the best (difference less than 0.015 in the correlation values and less than 5% in the deviation value) are highlighted with black dots. Each model run, resulting in a single data point for each of the evaluation metrics, comprises 138 NPF event days.**

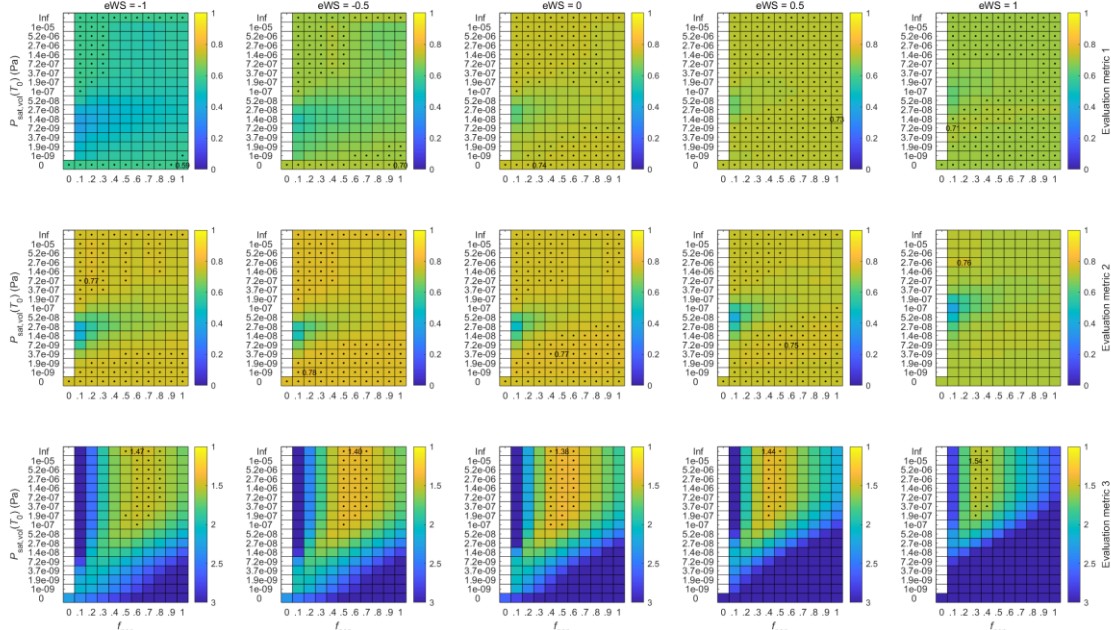

**Figure A4. Evaluation of the model performance with different exponents for the wind speed term (eWS) when eBLH = 0.5, eCS = 0, $f_{vol}$ = 1 and $\Delta h_{vol}$ = 80 kJ mol$^{-1}$. Each column of panels contains the performance matrices of the three different evaluation metrics as a function of the concentration multiplier for the non-volatile compound ($f_{non}$) and the saturation vapor pressure of the volatile compound ($P_{sat,vol}(T_0)$) with a specific eWS value. On the first two rows, higher correlation values of the evaluation metrics indicate better model performance, while on the last row, lower deviation correspond to better results. In each panel, the value of the best model performance is shown in numbers. Values close to the best (difference less than 0.015 in the correlation values and less than 5% in the deviation value) are highlighted with black dots. Each model run, resulting in a single data point for each of the evaluation metrics, comprises 138 NPF event days.**



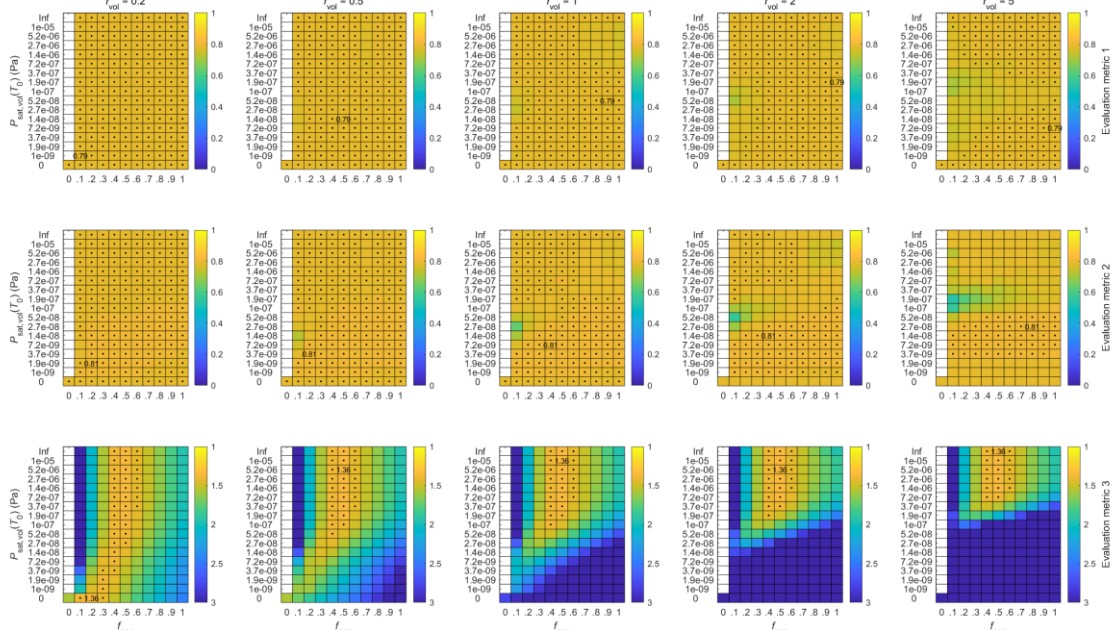

**Figure A5. Evaluation of the model performance with different concentration multipliers for the volatile component ($f_{vol}$) when eBLH = 0.5, eWS = 0, eCS = -0.5 and $\Delta h_{vol}$ = 80 kJ mol$^{-1}$. Each column of panels contains the performance matrices of the three different evaluation metrics as a function of the concentration multiplier for the non-volatile compound ($f_{non}$) and the saturation vapor pressure of the volatile compound ($P_{sat,vol}(T_0)$) with a specific $f_{vol}$ value. On the first two rows, higher correlation values of the evaluation metrics indicate better model performance, while on the last row, lower deviation correspond to better results. In each panel, the value of the best model performance is shown in numbers. Values close to the best (difference less than 0.015 in the correlation values and less than 5% in the deviation value) are highlighted with black dots. Each model run, resulting in a single data point for each of the evaluation metrics, comprises 138 NPF event days.**

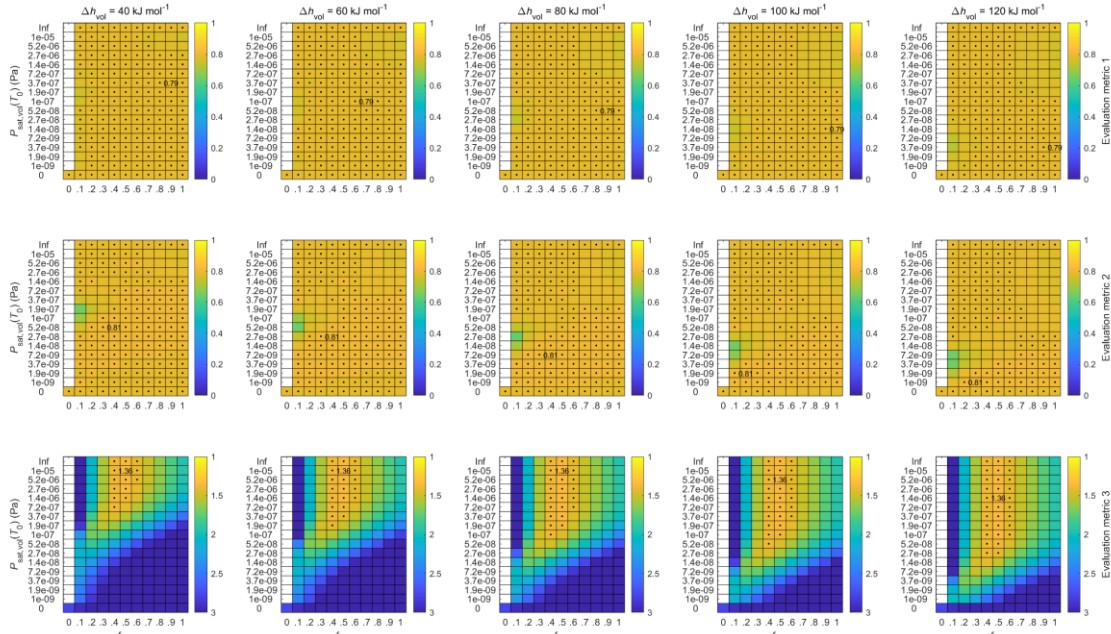

**Figure A6. Evaluation of the model performance with different enthalpies of vaporization for the volatile component ($\Delta h_{vol}$) when eBLH = 0.5, eWS = 0, eCS = -0.5 and $f_{vol}$ = 80 kJ mol$^{-1}$. Each column of panels contains the performance matrices of the three different evaluation metrics as a function of the concentration multiplier for the non-volatile compound ($f_{non}$) and the saturation vapor pressure of the volatile compound ($P_{sat,vol}(T_0)$) with a specific $\Delta h_{vol}$ value. On the first two rows, higher correlation values of the evaluation metrics indicate better model performance, while on the last row, lower deviation correspond to better results. In each panel, the value of the best model performance is shown in numbers. Values close to the best (difference less than 0.015 in the correlation values and less than 5% in the deviation value) are highlighted with black dots. Each model run, resulting in a single data point for each of the evaluation metrics, comprises 138 NPF event days.**

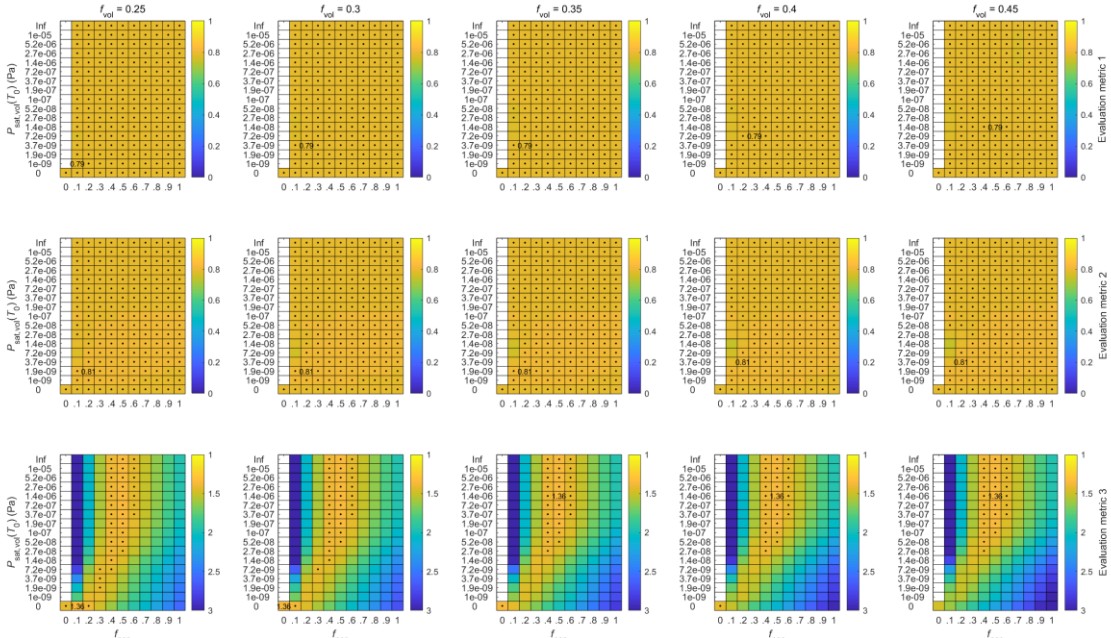

**Figure A7.** Evaluation of the model performance with different concentration multipliers for the volatile component ($f_{vol}$) in the 0.2-0.5 range when eBLH = 0.5, eWS = 0, eCS = -0.5 and $\Delta h_{vol}$ = 80 kJ mol$^{-1}$. Each column of panels contains the performance matrices of the three different evaluation metrics as a function of the concentration multiplier for the non-volatile compound ($f_{non}$) and the saturation vapor pressure of the volatile compound ($P_{sat,vol}(T_0)$) with a specific $f_{vol}$ value. On the first two rows, higher correlation values of the evaluation metrics indicate better model performance, while on the last row, lower deviation correspond to better results. In each panel, the value of the best model performance is shown in numbers. Values close to the best (difference less than 0.015 in the correlation values and less than 5% in the deviation value) are highlighted with black dots. Each model run, resulting in a single data point for each of the evaluation metrics, comprises 138 NPF event days.







**Figure A8: Same as Fig. 10 in the main text but with the horizontal axis in hours relative to the observation time of (a) the daily largest modeled particle ($D_{p,max}$) and (b) the smallest modeled particle in the evening ($D_{p,min}$) instead of the local time.**