# Peer review of "Explaining apparent particle shrinkage related to new particle formation events in western Saudi Arabia does not require evaporation"

_EGUsphere, 2023_

## Author Comment (AC1)

**REPLIES TO REFEREES**

We thank the referees for their comments and suggestions that have helped us improve our manuscript.
We have answered to each of the referee's comments below. The reviewers' comments are shown in **bold**, our replies in regular font and the text that has been added to the revised manuscript is shown in *red italics*. All changes in the revised manuscript are displayed using the Word 'Track changes' feature. The page and line numbers given in the answers refer to those in the ACPD version of the manuscript.

**Reply to Referee #1**

**This study uses statistical methods and model simulations in an attempt to explain the decreasing mode diameter (DMD) events that were preceded by new particle formation (NPF) in western Saudi Arabia. The study area has frequent NPF and DMD events, and thus the collected data are of special interest in the study of aerosol formation and growth, as well as apparent particle shrinkage. The manuscript is well organized and written with clear hypotheses, but could be shortened and improved by focusing on the major findings. The presented methodology is reasonable and includes new analytical approaches to elucidating the causes of DMD events. The presented results are explained in detail and adequately discussed with existing work. The summary and conclusions, however, could be more specific and quantitative. My specific comments are given below:**

We are pleased by the overall positive feedback on the manuscript. As the length of the manuscript was brought up by both the reviewers as well as the community comment, we wish to address it already here. During the preparation stage of the manuscript we also noticed that the manuscript was becoming quite lengthy. Because of this, we already experimented with several possible ways of shortening it but could not find a satisfactory solution. Due to the general lack of observational data and the resulting amount of assumptions made, we feel that it's important to be thorough and transparent with the discussion of the resulting limitations, uncertainties and interpretations of the results. While this lengthens the manuscript and potentially reduces its impact, we wish to give the reader enough material to evaluate the validity and robustness of the results without digging through supplementary files. We hope that the current way of presentation is still in line with the style of the journal.

1. **P4, L12, P11, L4-6, P14, L24: To put the observations in perspective, it would be helpful to provide the typical ground-level SO2 and PM concentrations at the study area and in Jeddah (or the region).**

We agree and added the following sentence on P4, L13:

*Measured $SO_2$ concentrations in the coastal area range from below 10 ppb to over 100 ppb (Al-Jeelani, 2009; Munir et al., 2013; Al-Jeelani, 2014; Ukhov et al., 2020b; Osipov et al., 2022) and the $PM_{2.5}$ varies between some tens and several hundreds of $\mu g\ m^{-3}$. The submicron particle mass ($PM_1$) is dominated by anthropogenic species and the coarse mode ($PM_{10}$) by mineral dust, while the $PM_{2.5}$ is significantly influenced by both (Khodeir et al., 2012; Lihavainen et al., 2016; Ukhov et al., 2020a; Osipov et al., 2022).*

We also rearranged the text in the referenced section (Sect. 2.1) for it to flow better with the added text.

**2. P7, Table 1: Why is the T0 set at 278K? Please consider giving the exact numbers or steps of Psat, instead of "…" Also, please replace "…" with "to" or "–" in the text.**

The selection of the reference temperature for the saturation vapor pressures is largely arbitrary, as long as the relevant volatility range in the ambient temperatures is covered (which it is, as shown in Sect. 4.3). Commonly used reference temperatures include at least 273 K, 278 K and 300 K, out of which we chose 278 K. We added a text in the caption of Table 1. that states how the Psat range is covered:

"The saturation vapor pressure range of $P_{sat,vol}(T_0) = $ 1e-9…1e-5 Pa *is studied using 15 logarithmically spaced values and it* corresponds to a saturation mass concentration range of …"

Using three dots to indicate a range of values is within the formatting guidelines of the journal (https://www.atmospheric-chemistry-and-physics.net/submission.html#math, last accessed 26.5.2023)

**3. If applicable, please consider providing relevant references to the presented equations.**

We added the reference Olenius et al. (2018) for Eq. (2) and Nieminen et al. (2010) for Eq. (5). We also removed the summation mark from Eq. (2), as the condensational growth rate is written for species k (and not for all species combined).

**4. P13, L5-6: Please be specific. Pearson correlation coeff.? Chi-square?**

We added text to specify that we're using Pearson's correlation coefficient in Evaluation metric 1 and the square root of the coefficient of determination in Evaluation metric 2.

**5. Fig. 2: This is a nice and interesting illustration of the stated hypotheses. It would be more informative if the typical transport characteristics could also be considered in the graph, e.g., the average wind speed during the DMD events, and how far into the sea at the onset of NFP. I suggest adding this info in the discussion sections (e.g., P17, L13, and Fig. 6).**

Thank you. We included more distance markers in Fig. 2 to clarify that the axis is to be interpreted as linear. With the particle positions at different times being specified by the colored markers, the typical wind speeds can be inferred from the figure (being on the order of 10 km h$^{-1}$ before and around noon and 20 km h$^{-1}$ in the later afternoon). We also increased the font size of the small texts in Fig. 2 and updated the colors. Additionally, minor clarifying edits were made to the caption.

A scale bar was also added to Fig. 6c to more clearly illustrate the distance scales in the displayed maps.

**6. P23, L9: Fig. 6 is informative. But what about including a plot of the probability of observing DMD particles? Would that be more straightforward for discussion?**

Thank you. While the same information is provided whether we show the probability of growth or DMD (since here $P_{growth} + P_{DMD} = 1$), we agree that it is better to explicitly display the probability

for DMD, since the DMD phase is the main focus of the paper. We replaced Fig. 6 and edited the caption and the related text (P23, L8-9) accordingly. We also changed the middle color in the colorbar of Fig. 6c to light grey in order to avoid confusion with regions where data is not displayed. We also clarified the meaning of the total NPF footprint in the caption.

7. **Section 4.3.1: The simulated outcomes of the model somewhat do not seem to capture the expected results or relationships, except wind speed. Please consider discussing the important variables that may be missing in the model. This is crucial for subsequent discussion about the model predictions in the following sections.**

On P11, L11-17 we do mention why we could expect a negative exponent for the CS term and discuss the limitations related to the formulation of this term. To clarify that the possible temporal variation in precursor emissions is not considered, and that along with the unknown vertical distribution of precursors this could affect the obtained exponents for BLH and WS, we added the following text to P11, L11:

*This is partly to account for the fact that the possible diurnal and seasonal variation in precursor emissions is not included in Eq. (7), which could counteract or change the signs of the exponents in case of similar variability. In addition, the response in precursor concentrations with respect to increasing BLH depends on the unknown vertical distribution of the precursors.*

8. **Discussion: Based on the observed and model results, is it possible that the DMD phenomena are a result of measurements at the edge (i.e., boundary) of the NPF event? In other words, the NPF event is analogous to a large-scale "plume" event, in which condensable vapors and hence particle growth is strongest in the middle of the plume (i.e., source region), and decreases to the lowest at the edge (consider a plume traveling over a fixed measurement site).**

We are not sure if we completely understand the referred situation/analogy. We do not believe that the DMD phenomenon is related to measuring directly *at* the edge/boundary of the NPF event but that the crucial thing is measuring at a location where air masses *from* (and beyond) such boundaries (concentration gradients) are observed. The presence of such a near-by boundary, which leads to spatial differences in the concentrations of condensable vapors, and hence particle growth, is a prerequisite for the apparent shrinkage process. We also think that there is definitely a plume of sulfate aerosol created which passes over the measurement site, but the main reason why we observe the low sulfate "tail" of this plume is that once the sulfate production starts to decrease with decreasing photochemistry, less and less sulfate (and smaller particles) will be formed in the air masses travelling over the strong SO2 emission areas. Such eastward traveling plumes of sulfate can also be seen in the beautiful animation created from the WRF-Chem simulations by Osipov et al. (2022) (https://www.youtube.com/watch?v=KF12n3gJxjU, last accessed 28.5.2023).

Whether there is a plume of SO2 (and other precursors) traveling over the measurement site is, however, much less clear. Our base assumption with using the temporally constant satellite-derived SO2 field is that the emissions of SO2 are relatively constant over the diurnal cycle (could apply to mainly industrial sources) and that no SO2 plume is created. Looking e.g. at the NASA/GMAO GEOS forecast simulations of the surface SO2 in the region (https://fluid.nccs.nasa.gov/wxmaps/chem2d/, last accessed 29.5.2023) suggest that while the SO2 field shows some diurnal variation depending on the prevailing winds etc., the highest concentrations remain over the main source areas and no distinct plumes are created. Of course, the

diurnal variation in emissions might also not be realistic in these simulations. In our model results, we find the best performance with the positive BLH exponent, which suggest that there might be an emission peak during the afternoon, which again could create an SO2 plume. In addition, the accumulation of SO2 during the night over the main source areas (due to calm conditions and reduced photochemical loss) could also create an eastward traveling SO2 plume during the day. However, our model is unable to describe the propagation of such precursor plumes (e.g. in the case of the BLH dependency, the higher concentration with higher BLH is applied regardless of the air mass position with respect to the emission sources). Nevertheless, we believe that the very strong diurnal variation in the photochemical production of condensable vapors dominates over the possible diurnal variation in the precursor vapors. This is supported by the relatively good model performance when only the temporally constant SO2 field and radiation is included (P35, L13-22 and Fig. A3).

9. **Summary and conclusions: It would be helpful to provide the specific boundary conditions ([SO2], CS, WS, WD, T, BLH, Cvol, Cnon, etc.) that favor the occurrence of DMD in the study area. These could be used as a reference for other studies.**

In the discussion section, the paragraphs starting at P34, L12 and P34, L21 go through the main factors contributing to favorable conditions for DMD in the study region and comment on the occurrence of similar conditions on other sites. The only specific "boundary condition" for the apparent shrinkage is the presence of distinct spatial variation in aerosol growth rates (which is likely most commonly caused by variation in aerosol precursor concentrations). To highlight this also in the summary and conclusions and to give a clear reference for other studies we added the following sentence on P37, L13:

*The main enabling factor for the apparent shrinkage is the distinct gradient in the aerosol precursor concentrations near the measurement location.*

and the following sentences on P37, L22:

*Spatially varying growth rates, which enable the apparent shrinkage, could occur in many environments especially due to the emission differences between land and sea areas, as well as those between urban and natural areas. To give a rough guideline, if a notable decrease in aerosol precursor concentrations can be expected approximately within the distance covered by the mean wind from the onset of NPF to the onset of DMD (in our case ~ 10 km h$^{-1}$ × 6 h = 60 km), the apparent shrinkage process should be considered as a likely candidate for the DMD. If the precursor source distribution varies in different directions from the measurement site, the occurrence of DMD with wind direction shifting away from stronger source areas could also be indicative of the apparent shrinkage.*

We also modified the rest of the concluding paragraph to fit better with the added text.

**Reply to Referee #2**

**The MS deals with decreasing mode diameter NPF (DMD) events in a special atmospheric environment. These events are important to formulate and check ideas on the volatility properties of the chemical species involved in the particle growth. The special environment is characterised by a very high relative occurrence frequency of NPF events of 73% without evident seasonal tendency. Of them 76% were identified as DMD events. These characteristics are related to the local properties at the site. The MS covers a complex study with a nice adaptation of physicochemical approaches. The findings are valuable and are discussed in detail. It is relevant and of interest for the scientific community. I can definitely propose its acceptance. The following comments can be advised to the authors for consideration.**

We are pleased by the overall positive feedback. Thank you.

**1. The MS seems to be long and dense. The authors may want to reorganise it in a way that it focuses in the main text on the main relationships, on the conditions for their validity, on discussions of the bounding conditions and on their physical meaning, and on the findings, results and conclusions. The detailed derivation of the relationships could be in the Appendix. In this way, the authors would better communicate the messages. The Summary and conclusions section should be shortened to primary messages.**

See response to the overview by Referee #1. Since we opt not to shorten the main text of the manuscript, we also feel that it's beneficial for both the Abstract and the Summary and conclusions sections to give a short outline of the whole study. This way, the reader who does not want to commit to going through the whole manuscript can get a reasonable idea of the study by only going through the summarizing sections. The conclusions were sharpened to more clearly address the relevance and implications of our study for other environments (response to comment 9 by Referee #1).

**2. The MS could be reorganised also from structural point of view. The part P13, L26 – P14, L9 belongs more to the site description than to Hypotheses based on our previous results. Further parts (for instance P4, L6) are redundant in an article.**

We agree that the placement of the part on P13, L26 – P14, L9 is slightly awkward and that this information could be provided either already in the introduction or in the site description. However, since the part discusses the classification of NPF events and the different phases, which are described after the site description in the measurements and methods, and since the presented previous results are related to the hypothesized causes, we still felt that the current place is the most suitable.

We also agree that the part on P4, L6 is redundant and removed it.

**3. The actual results were obtained under very specific atmospheric and geographic conditions. Could the spatially localised (local) nucleating plume affect its dynamic development? In that sense, the MS could advantageous be complemented with a brief discussion on the representativity of its finding. Could they be valid for other, more ordinary locations?**

This comment concerns similar issues as those raised by Referee #1 in comments no. 8 and 9, and we refer to the answer provided there. In short, a brief discussion on the representativity is already

provided and the suggested apparent shrinkage can occur anywhere, where aerosol growth rate shows spatial variability. The most notable causes for such variability include the differences between land and sea areas, as well as those between urban and natural areas. Therefore, the required conditions are actually not very specific but several factors at our study site in Hada Al Sham do contribute favorably to the occurrence of these conditions.

**4. Paragraph P37, L15-19 are not completely clear. Reformulation is requested.**

It would be helpful if the reviewer clarified what is the source of confusion on the referenced paragraph or how they understand it. What we wish to state is that if the particles formed in NPF exceed some CCN-size threshold (e.g. 100 nm) only momentarily before evaporating back to smaller sizes, the net production of CCN-sized particles from NPF is essentially zero. However, if no evaporation occurs (as our results suggest), all the particles that we observe to exceed the CCN-size threshold will also remain in these larger sizes. In addition, our results suggest the observed particles are in reality constantly growing (despite the DMD), which means that the smaller particles observed during the DMD phase (that are not above the CCN-threshold) will actually continue their growth towards larger sizes as they are transported beyond our observations, and might thus reach the CCN-threshold later in time. This clearly increases the relevance of NPF towards CCN in this region compared to the case of evaporating particles. We feel that this is quite clearly stated in the referenced paragraph and found no obvious sources of confusion or mistakes.

**5. Formal comments. Altitude is missing in P4, L10. Insert space after semicolons in many places (e.g. in references and P4, L16). Internal brackets in P4, L17 are redundant, while semicolon is needed.**

We added the altitude information and reformatted the references.
* * *
In addition to the changes suggested by the referees, we made several minor adjustments to the text which are all displayed using the 'Track changes' feature in the revised manuscript. The most notable changes are:

P4, L3: we added text specifying the more general use-case of the developed methods

"Similar methods to those developed and applied in this study could also be used on other sites to study the cause of DMD events *or the role of transport effects in aerosol growth analysis in general*."

P10, L25: We added a sentence supporting our assumption of the spatial distribution of the precursor for the potentially volatile component

*This is supported by a similar spatial distribution of the satellite-retrieved formaldehyde (De Smedt et al., 2021), which is an intermediate gas in the oxidation chains of VOCs.*

P15, L20: We specify higher evaporation rate of the condensable vapors separately from decreased concentration since increasing volatility does not reduce the concentration of condensable vapors (as previously implied).

P15, L23: We added the possible causes to condition 2b to match the formulation of 2a.

We also wish to state that we are aware that the red and green shadings in Figs. 3, 4 and 5 might not be distinguishable for readers with color vision deficiencies but note that this is not essential for the correct interpretation of the findings.

**References:**

Al-Jeelani, H. A.: Diurnal and Seasonal Variations of Surface Ozone and Its Precursors in the Atmosphere of Yanbu, Saudi Arabia %J Journal of Environmental Protection, Vol.05No.05, 15, 10.4236/jep.2014.55044, 2014.

Al-Jeelani, H. A. J. T. J. o. A. S. R.: Evaluation of air quality in the Holy Makkah during Hajj season 1425 H, 115-121, 2009.

De Smedt, I., Pinardi, G., Vigouroux, C., Compernolle, S., Bais, A., Benavent, N., Boersma, F., Chan, K. L., Donner, S., Eichmann, K. U., Hedelt, P., Hendrick, F., Irie, H., Kumar, V., Lambert, J. C., Langerock, B., Lerot, C., Liu, C., Loyola, D., Piters, A., Richter, A., Rivera Cárdenas, C., Romahn, F., Ryan, R. G., Sinha, V., Theys, N., Vlietinck, J., Wagner, T., Wang, T., Yu, H., and Van Roozendael, M.: Comparative assessment of TROPOMI and OMI formaldehyde observations and validation against MAX-DOAS network column measurements, Atmos. Chem. Phys., 21, 12561-12593, 10.5194/acp-21-12561-2021, 2021.

Khodeir, M., Shamy, M., Alghamdi, M., Zhong, M., Sun, H., Costa, M., Chen, L.-C., and Maciejczyk, P.: Source apportionment and elemental composition of PM2.5 and PM10 in Jeddah City, Saudi Arabia, Atmospheric Pollution Research, 3, 331-340, https://doi.org/10.5094/APR.2012.037, 2012.

Lihavainen, H., Alghamdi, M. A., Hyvarinen, A. P., Hussein, T., Aaltonen, V., Abdelmaksoud, A. S., Al-Jeelani, H., Almazroui, M., Almehmadi, F. M., Al Zawad, F. M., Hakala, J., Khoder, M., Neitola, K., Petaja, T., Shabbaj, I. I., and Hameri, K.: Aerosols physical properties at Hada Al Sham, western Saudi Arabia, Atmospheric Environment, 135, 109-117, 10.1016/j.atmosenv.2016.04.001, 2016.

Munir, S., Habeebullah, T. M., Seroji, A. R., Morsy, E. A., Mohammed, A. M. F., Saud, W. A., Abdou, A. E. A., and Awad, A. H.: Modeling Particulate Matter Concentrations in Makkah, Applying a Statistical Modeling Approach, Aerosol and Air Quality Research, 13, 901-910, 10.4209/aaqr.2012.11.0314, 2013.

Nieminen, T., Lehtinen, K. E. J., and Kulmala, M.: Sub-10 nm particle growth by vapor condensation - effects of vapor molecule size and particle thermal speed, Atmospheric Chemistry and Physics, 10, 9773-9779, 10.5194/acp-10-9773-2010, 2010.

Olenius, T., Pichelstorfer, L., Stolzenburg, D., Winkler, P. M., Lehtinen, K. E. J., and Riipinen, I.: Robust metric for quantifying the importance of stochastic effects on nanoparticle growth, Sci Rep-Uk, 8, 14160, 10.1038/s41598-018-32610-z, 2018.

Osipov, S., Chowdhury, S., Crowley, J. N., Tadic, I., Drewnick, F., Borrmann, S., Eger, P., Fachinger, F., Fischer, H., Predybaylo, E., Fnais, M., Harder, H., Pikridas, M., Vouterakos, P., Pozzer, A., Sciare, J., Ukhov, A., Stenchikov, G. L., Williams, J., and Lelieveld, J.: Severe atmospheric pollution in the Middle East is attributable to anthropogenic sources, Communications Earth & Environment, 3, 203, 10.1038/s43247-022-00514-6, 2022.

Ukhov, A., Mostamandi, S., da Silva, A., Flemming, J., Alshehri, Y., Shevchenko, I., and Stenchikov, G.: Assessment of natural and anthropogenic aerosol air pollution in the Middle East using MERRA-2, CAMS data assimilation products, and high-resolution WRF-Chem model simulations, Atmos. Chem. Phys., 20, 9281-9310, 10.5194/acp-20-9281-2020, 2020a.

Ukhov, A., Mostamandi, S., Krotkov, N., Flemming, J., da Silva, A., Li, C., Fioletov, V., McLinden, C., Anisimov, A., Alshehri, Y. M., and Stenchikov, G.: Study of SO Pollution in the Middle East Using MERRA-2, CAMS Data Assimilation Products, and High-Resolution WRF-Chem Simulations, Journal of Geophysical Research: Atmospheres, 125, e2019JD031993, https://doi.org/10.1029/2019JD031993, 2020b.